# SpaFL: Communication-Efficient Federated Learning with Sparse Models and Low Computational Overhead

**Minsu Kim** [*]
Virginia Tech

**Walid Saad**
Virginia Tech

**Merouane Debbah**
Khalifa University

**Choong Seon Hong**
Kyung Hee University

## Abstract

The large communication and computation overhead of federated learning (FL) is one of the main challenges facing its practical deployment over resource-constrained clients and systems. In this work, SpaFL: a communication-efficient FL framework is proposed to optimize sparse model structures with low computational overhead. In SpaFL, a trainable threshold is defined for each filter/neuron to prune its all connected parameters, thereby leading to structured sparsity. To optimize the pruning process itself, only thresholds are communicated between a server and clients instead of parameters, thereby learning how to prune. Further, global thresholds are used to update model parameters by extracting aggregated parameter importance. The generalization bound of SpaFL is also derived, thereby proving key insights on the relation between sparsity and performance. Experimental results show that SpaFL improves accuracy while requiring much less communication and computing resources compared to sparse baselines. The code is available at https://github.com/news-vt/SpaFL_NeruIPS_2024

## 1 Introduction

Federated learning (FL) is a distributed machine learning framework in which clients collaborate to train a machine learning (ML) model without sharing private data [1]. In FL, clients perform multiple epochs of local training using their own datasets and communicate model updates with a server. Different from a classical, centralized ML, FL systems are typically deployed on edge devices such as mobile or Internet of Things (IoT) devices, which have limited computing and communication resources. However, current ML models are typically too large and complex to be trained and deployed for inference by edge devices. Moreover, large model sizes can induce significant FL communication costs on both devices and communication networks. Hence, the practical deployment of FL over *resource-constrained devices and systems* requires optimized computation and communication costs for both edge devices and communication networks. This has motivated lines of research focused on reducing communication overhead in FL [2, 3], training sparse models in FL [4, 5, 6, 7, 8, 9], and optimizing model architectures to find a compact model for inference [10, 11, 12]. The works in [2, 3] proposed training algorithms such as quantization, gradient compression, and transmitting the subset of models in order to reduce the communication costs of FL. However, the associated computational overhead of these existing algorithms remains high since devices have to train a dense model. In [4, 5, 6, 7, 8, 9], FL algorithms in which devices train and communicate sparse models are proposed. However, the works in [4, 5] used unstructured pruning, which is difficult to gain the computation efficiency in practice. Moreover, the computation

---

[*]This work was supported in part by a grant from the Amazon-Virginia Tech Initiative for Efficient and in part by the Robust Machine Learning and U.S. National Science Foundation under Grant CNS-2114267.

and communication overhead can still be large if model sparsity is not high. In [6, 7, 8, 9], the authors investigated the structured sparsity, however, the solutions therein either fixed the channel sparsity patterns for clients or did not optimize the pruning process. Furthermore, the FL approaches of [10, 11, 12] can significantly increase computation resource usage by training multiple models for resource-constrained devices. Clearly, despite a surge of literature on sparsity in FL, there is still a need to develop new FL algorithms that can find sparse model structures with optimized communication efficiency and low computational overhead to operate on resource-constrained devices.

The main contribution of this paper is *SpaFL: a communication-efficient FL framework for optimizing sparse models with low computational overhead* achieved by performing structured pruning through trainable thresholds. Here, a trainable threshold is defined for each filter/neuron to prune all of its connected parameters. To optimize the pruning process, *only thresholds are communicated* between clients and the FL server. Hence, clients can learn how to prune their model from global thresholds and can significantly reduce communication costs. Since parameters are not communicated, the clients' parameters and sparse model structures will remain personalized while only global thresholds are shared. We show that global thresholds can capture the aggregated parameter importance of clients. We further update the clients' model parameters by extracting aggregated parameter importance from global thresholds to improve performance. We analyze the generalization ability of SpaFL and provide insights on the relation between sparsity and performance. We summarize our contributions as follows:

- We propose a new communication-efficient FL framework called SpaFL, in which clients optimize their sparse model structures with low computing costs through trainable thresholds.

- We show how SpaFL can significantly reduce communication overhead for both clients and the server by only exchanging thresholds, the number of which is less than two orders of magnitude smaller than the number of model parameters.

- We provide the generalization performance of SpaFL. Moreover, the impact of sharing thresholds on the model performance is theoretically and experimentally analyzed.

- Experimental results demonstrate the performance, computation costs, and communication efficiency of SpaFL compared with both dense and sparse baselines. For instance, the results show that SpaFL uses only 0.17% of communication and 12.0% of computation resources compared to a dense baseline FedAvg while improving accuracy. Additionally, SpaFL improves accuracy by 2.92% compared to a sparse baseline while consuming only 0.35% of this baseline's communication resources, and only 24% of its computing resources.

## 2 Background and Related Work

### 2.1 Federated Learning

Distributed machine learning has consistently progressed and achieved success. However, it mostly focuses on training with independent and identically distributed (i.i.d.) data [13, 14]. The FL frameworks along with the FedAvg [1] enables clients to collaboratively train while preserving data privacy without data sharing. Due to privacy constraints and individual preferences, FL clients often collect non-iid data. As such, data can exhibit differences and imbalances in distribution across clients. This variability poses significant challenges in achieving efficient convergence. For a more detailed literature review, we refer to [15, 16]. Although most of state-of-the-art FL methods are effective in mitigating data heterogeneity, they often neglect the computational and communication costs involved in the training process.

### 2.2 Training and Finding Sparse Models in FL

To reduce the computation and communication overhead of complex ML models during training, the idea of embedding FL algorithms with pruning has recently attracted attention. In [4, 5, 6, 7, 8, 9, 17, 18, 19, 20, 21, 22, 23, 24, 25], the clients train sparse models and communicate sparse model parameters to reduce computation and communication overhead. To improve the aggregation phase with sparse models, the works in [17, 20, 21] perform averaging only between overlapping parameters to avoid information dilution by excluding zero value parameters. The authors in [18] obtained a sparse model by selecting a particular client to prune an initial dense model and then

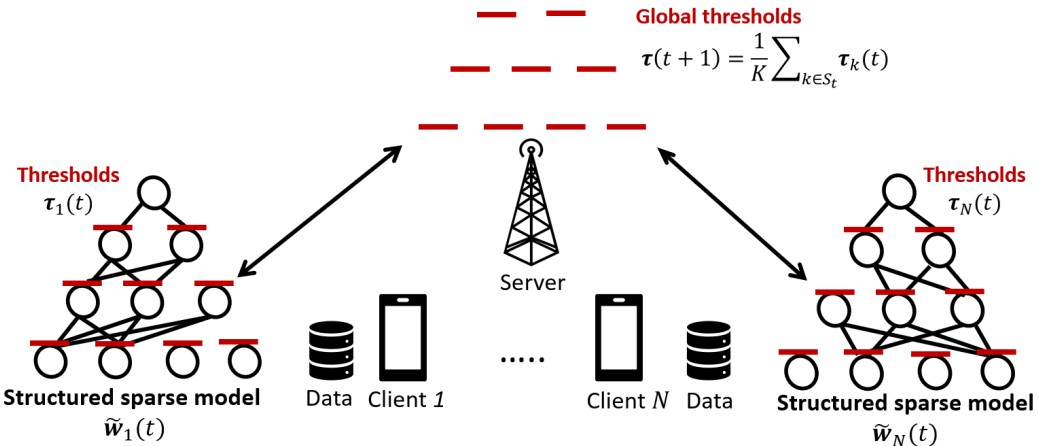

Figure 1: Illustration of SpaFL framework that performs model pruning through thresholds. Only the thresholds are communicated between the server and clients.

performed training in a similar way to FedAvg. In [4, 24], the authors presented binary masks adjustment strategy to improve the performance of sparse models and communication efficiency. The work in [25] progressively pruned a dense model for sparsification and analyzed its convergence. In [19, 22], the clients optimized personalized sparse models by exchanging lottery tickets [26] at every communication round. The work in [5] obtained personalized sparse models by $l_1$ norms constraints and the correlation between local and global models. In [8], the authors proposed dual pruning scheme for both local and global models to reduce the communication costs. The FL framework of [23] allows clients to train personalized sparse models in a decentralized setting without a central server. Although these works [17, 18, 4, 24, 25, 19, 22, 5, 23] adopted sparse models during training, they used unstructured pruning, which is difficult to improve the computation efficiency in practice. Meanwhile, with structured sparsity, the authors [7] proposed a training scheme that allows clients to train smaller submodels of a global model. In [9], clients train set of submodels with fixed channel sparsity patterns depending on their computing capabilities. The work in [6] studied structured sparsity by adjusting clients' channel activation probabilities. However, the works in [7, 9] fixed sparsity patterns and did not optimize sparse model structures. Although [6] optimized channel activation probabilities, the communication cost of downlink still remains high as a server broadcasts whole parameters. Similar to our work, in [27, 28], only binary masks are communicated and optimized by training auxiliary variables to learn sparse model structures. However, the work in [27] approximated binarization step using a sigmoid function during forward propagation. In [28], the downlink communication costs remained the same as that of FedAvg. In [10, 11, 29], clients perform neural-architecture-search by training multiple models to find optimized and sparse models to improve computational and memory efficiency at inference phase. However, in practice, clients often have limited resources to support the computationally intensive architecture search process [30]. Therefore, most prior works either adopted unstructured pruning or they still required extensive computing and communication costs for finding optimal sparse models. In contrast to the prior art, in the proposed SpaFL framework, we find sparse model structures with structured sparsity by optimizing and communicating trainable thresholds for filter/neurons.

## 3 SpaFL Algorithm

In this section, we first present the proposed pruning scheme for structured sparsity and formulate our FL problem to find optimal sparse models. Then, we present SpaFL to solve the proposed problem with low computation and communication overhead.

### 3.1 Structured Pruning with Trainable Thresholds

We define a trainable threshold for each neuron in linear layers or for each filter in convolutional layers. The neural network of client $k$ will consist of $L$ layers as $\{\boldsymbol{W}_k^1, \ldots, \boldsymbol{W}_k^L\}$. For parameters $\boldsymbol{W}_k^l \in \mathbb{R}^{n_{\text{out}}^l \times n_{\text{in}}^l}$ in a linear layer $l$, we define trainable thresholds $\boldsymbol{\tau}^l \in \mathbb{R}^{n_{\text{out}}^l}$ for output neurons. If it

is a convolutional layer $\boldsymbol{W}_k^l \in \mathbb{R}^{n_{\text{out}}^l \times c_{\text{in}}^l \times k^l \times h^l}$, where $c_{\text{in}}^l$ is the number of input channels and $k^l \times h^l$ are the kernel sizes, we can change $\boldsymbol{W}_k^l$ as $\boldsymbol{W}_k^l \in \mathbb{R}^{n_{\text{out}}^l \times n_{\text{in}}^l}$ with $n_{\text{in}}^l = c_{\text{in}}^l \times k^l \times h^l$. Similarly, we can define the corresponding thresholds $\boldsymbol{\tau}^l \in \mathbb{R}^{n_{\text{out}}^l}$ for filters in that layer. Then, for each client $k$, we define a set of total thresholds $\boldsymbol{\tau} = \{\boldsymbol{\tau}^1, \ldots, \boldsymbol{\tau}^L\}$. Note that the number of these additional thresholds will be at most 1% of the number of model parameters $d$.

For threshold $\boldsymbol{\tau}_i^l$ of filter/neuron $i$ in layer $l$, we compare the average magnitude of its connected parameters $\mu_{k,i}^l = 1/n_{\text{in}}^l \sum_{j=1}^{n_{\text{in}}^l} |w_{k,ij}^l|$ to its value $\boldsymbol{\tau}_i^l$. If $\mu_{k,i}^l < \boldsymbol{\tau}_i^l$, we prune all connected parameters to this filter/neuron. Hence, our pruning can induce structured sparsity unlike [31]. Thus, we do not need to compute the gradients of parameters in a pruned filter/neuron [32] during backpropagation. We can obtain a binary mask $\boldsymbol{p}_k^l$ for $\boldsymbol{W}_k^l$, as follows

$$p_{k,ij}^l = S(\mu_{k,i} - \tau_i^l), \ 1 \leq i \leq n_{\text{out}}^l, 1 \leq j \leq n_{\text{in}}^l, \tag{1}$$

where $S(\cdot)$ is a unit step function. Hence, we can obtain the binary masks $\{\boldsymbol{p}_k^1, \ldots, \boldsymbol{p}_k^L\}$ by performing (1) at each layer. To facilitate the pruning, we constrain the parameters and thresholds to be within $[-1, 1]$ and $[0, 1]$, respectively [31]. For simplicity, we unroll $\{\boldsymbol{W}_k^1, \ldots, \boldsymbol{W}_k^L\}$ and $\{\boldsymbol{p}_k^1, \ldots, \boldsymbol{p}_k^L\}$ to $\boldsymbol{w}_k \in \mathbb{R}^d$ and $\boldsymbol{p}_k \in \mathbb{R}^d$, respectively as done in [33]. Thresholds represent the importance of their connected parameters (see more details in Section 3.3.1). Hence, clients can know which filter/neuron is important by training thresholds, thereby optimizing sparse model structures. Then, the key question becomes: Can clients benefit by collaborating to optimize shared thresholds in order to find optimal sparse models? We partially answer this question in Table 1. Following the same configurations in Section 5, clients with non-iid datasets only train and communicate thresholds $\boldsymbol{\tau}$ while freezing model parameters.

| Algorithm | FMNIST | CIFAR-10 | CIFAR-100 |
|---|---|---|---|
| Trained $\boldsymbol{\tau}$ | **65.52±5.3** | **60.94±3.4** | **24.80±1.1** |
| Initialization | $10.22 \pm 0.25$ | $10.38 \pm 0.42$ | $1.43 \pm 0.53$ |

Table 1: Only thresholds are trained and communicated while parameters are kept frozen.

We can see that learning sparse structures can improve the performance even without training parameters. This also corroborates the result of [28]. Motivated by this observation, we aim to find optimal sparse models of clients in an FL setting by communicating only thresholds in order to reduce the communication costs in both clients and server sides while keeping parameters locally. The communication cost will decrease drastically because the number of thresholds will be at most 1% of the number of model parameters $d$. Essentially, we optimize the sparse models of clients with small computing and communication resources by communicating thresholds.

## 3.2 Problem Formulation

We aim to optimize each client's model parameters and sparse model structures jointly in a personalized FL setting by only communicating thresholds. This can be formulated as the following optimization problem:

$$\min_{\boldsymbol{\tau}, \boldsymbol{w}_1, \ldots, \boldsymbol{w}_N} \quad \frac{1}{N} \sum_{k=1}^N F_k(\tilde{\boldsymbol{w}}_k, \boldsymbol{\tau}),$$

$$\text{s.t.} \quad F_k(\tilde{\boldsymbol{w}}_k, \boldsymbol{\tau}) = \frac{1}{D_k} \sum_{i=1}^{D_k} \mathcal{L}(\boldsymbol{w}_k \odot \boldsymbol{p}_k(\boldsymbol{\tau}); \{\boldsymbol{x_i}, y_i\}), \tag{2}$$

where $\tilde{\boldsymbol{w}}_k = \boldsymbol{w}_k \odot \boldsymbol{p}_k(\boldsymbol{\tau})$ is a pruned model, $F_k(\cdot)$ is a empirical risk associated with local data of client $k$, $\mathcal{L}$ is a loss function, $D_k$ is the number of data samples, $\{\boldsymbol{x}, y\}$ is an input-label pair, $\boldsymbol{w}_k$ captures the model parameters, and $\odot$ is the Hadamard product. If an element of $\boldsymbol{p}_k(\boldsymbol{\tau})$ is zero, then the corresponding parameter of $\boldsymbol{w}_k$ will be pruned. Our goal is to obtain the optimal $\boldsymbol{w}_k$ and $\boldsymbol{\tau}$ for each client in order to reduce the computation and communication overhead during training. However, solving (2) is not trivial because $\boldsymbol{w}_k$ and $\boldsymbol{\tau}$ are highly correlated. Moreover, structured sparsity can induce a large performance drop due to coarse-grained sparsity patterns compared to unstructured pruning [34].

### 3.3 Algorithm Overview

We now describe the proposed algorithm, SpaFL, that can solve (2) while maintaining communication-efficiency with low computational cost. In SpaFL, every client jointly optimizes its personalized sparse model structure and model parameters with trainable thresholds, which can be used to prune filters/neurons. To save communication resources, only thresholds will be aggregated at a server to generate global thresholds for the next round. Here, global thresholds can represent the aggregated parameter importance of clients. Hence, at the beginning of each round, every client extracts the aggregated parameter importance from the global thresholds so as to update its model parameters. The overall algorithm is illustrated in Fig 1. and summarized in Algorithm 1.

#### 3.3.1 Local Training for Parameters and Thresholds

At each round, a server samples a set of clients $\mathcal{S}_t$ such that $|\mathcal{S}_t| = K$ for local training. For given global thresholds $\boldsymbol{\tau}(t)$ at round $t$, client $k \in \mathcal{S}_t$ generates a binary mask $\boldsymbol{p}_k(\boldsymbol{\tau}(t))$ using (1). Subsequently, it obtains the sparse model $\tilde{\boldsymbol{w}}_k(t) = \boldsymbol{w}_k(t) \odot \boldsymbol{p}_k(\boldsymbol{\tau}(t))$. To improve the communication efficiency, each sampled client performs $E$ epochs using mini-batch stochastic gradient to update parameters and thresholds as follows:

$$\boldsymbol{w}_k^{e+1}(t) \leftarrow \boldsymbol{w}_k^e(t) - \eta(t)\boldsymbol{g}_k(\tilde{\boldsymbol{w}}_k^e(t)), \ \tilde{\boldsymbol{w}}_k^0(t) = \tilde{\boldsymbol{w}}_k(t), \ 0 \le e \le E-1, \tag{3}$$

$$\boldsymbol{\tau}_k^{e+1}(t) \leftarrow \boldsymbol{\tau}_k^e(t) - \eta(t)\boldsymbol{h}_k(\tilde{\boldsymbol{w}}_k^e(t)), \ \boldsymbol{\tau}_k^0(t) = \boldsymbol{\tau}(t), \ 0 \le e \le E-1, \tag{4}$$

where $\boldsymbol{g}_k(\tilde{\boldsymbol{w}}_k^e(t)) = \nabla_{\tilde{\boldsymbol{w}}_k^e} F_k(\tilde{\boldsymbol{w}}_k^e(t), \boldsymbol{\tau}(t); \xi_k^e(t))$, $\boldsymbol{h}_k(\tilde{\boldsymbol{w}}_k^e(t)) = \nabla_{\boldsymbol{\tau}} F_k(\tilde{\boldsymbol{w}}_k^e(t), \boldsymbol{\tau}(t); \xi_k^e(t))$ with a mini-batch $\xi$ and $\eta(t)$ is a learning rate. Parameters of unpruned filter/neurons and thresholds will be jointly updated via backpropagation. To enforce sparsity, we add a regularization term $R(t)$ to (4) in order to penalize small threshold values. To this end, client $k$ first calculates the following sparsity regularization term $R(t) = \sum_{l=1}^{L} \sum_{i=1}^{n_{\text{out}}^l} \exp(-\tau_i)$. Then, the loss function can be rewritten as:

$$F_k(\tilde{\boldsymbol{w}}_k^e(t), \boldsymbol{\tau}(t); \xi_k^e(t)) \leftarrow F_k(\tilde{\boldsymbol{w}}_k^e(t), \boldsymbol{\tau}(t); \xi_k^e(t)) + \alpha R(t), \tag{5}$$

where $0 \le \alpha \le 1$ is the coefficient that controls $R(t)$. From (5), we can give thresholds $\boldsymbol{\tau}(t)$ performance feedback on the current sparse model while also progressively increasing $\boldsymbol{\tau}(t)$ through the sparsity regularization term $R(t)$ [31]. From (5), client $k$ then updates the received global thresholds $\boldsymbol{\tau}(t)$ via backpropagation as follows

$$\boldsymbol{\tau}_k^{e+1}(t) \leftarrow \boldsymbol{\tau}_k^e(t) - \eta(t)\boldsymbol{h}_k(\tilde{\boldsymbol{w}}_k^e(t)) + \alpha\eta(t)\exp\{-\boldsymbol{\tau}_k^e(t)\}. \tag{6}$$

After local training, each client $k \in \mathcal{S}_t$, transmits the updated thresholds $\boldsymbol{\tau}_k(t)$ to the server. Here, the communication overhead will be less than one percent of that of transmitting the entire parameters. Subsequently, the server performs aggregation and broadcasts new global thresholds, i.e.,

$$\boldsymbol{\tau}(t+1) = \frac{1}{K} \sum_{k \in \mathcal{S}_t} \boldsymbol{\tau}_k(t). \tag{7}$$

Here, in SpaFL, clients communicate only thresholds. Then, what will clients learn from sharing trained thresholds? Next, we show that thresholds represent the importance of their associated filter/neurons.

### 3.3.2 Learning Parameter Importance From Thresholds

Clients can know which filter/neurons are important by sharing trained thresholds. For the threshold of filter/neuron $i$ at layer $l$ of client $k$, its gradient can be written as below

$$h_{k,i}^l(\tilde{\boldsymbol{w}}_k^e(t)) = \frac{F_k(\tilde{\boldsymbol{w}}_k^e(t))}{\partial \tau_{k,i}^{e,l}(t)} = \sum_{j=1}^{n_{\text{in}}^l} \frac{\partial \tilde{w}_{k,ij}^{e,l}(t)}{\partial \tau_{k,i}^{e,l}(t)} \frac{\partial F_k(\tilde{\boldsymbol{w}}_k(t), \boldsymbol{\tau}(t))}{\partial \tilde{w}_{k,ij}^{e,l}(t)} = \sum_{j=1}^{n_{\text{in}}^l} \frac{\partial \tilde{w}_{k,ij}^{e,l}(t)}{\partial \tau_{k,i}^{e,l}(t)} \{\boldsymbol{g}_k(\tilde{\boldsymbol{w}}_k^e(t))\}_{ij}^l$$

$$= \sum_{j=1}^{n_{\text{in}}^l} \frac{\partial \tilde{w}_{k,ij}^{e,l}(t)}{\partial Q_{k,i}^{e,l}(t)} \frac{\partial Q_{k,i}^{e,l}(t)}{\partial \tau_{k,i}^{e,l}(t)} \{\boldsymbol{g}_k(\tilde{\boldsymbol{w}}_k^e(t))\}_{ij}^l$$

$$= \sum_{j=1}^{n_{\text{in}}^l} \frac{\partial w_{k,ij}^{e,l}(t) \odot p_{k,ij}^{e,l}(t)}{\partial S(Q_{k,i}^{e,l}(t))} \frac{\partial S(Q_{k,i}^{e,l}(t))}{Q_{k,i}^{e,l}(t)} \frac{\partial Q_{k,i}^{e,l}(t)}{\partial \tau_{k,i}^{e,l}(t)} \{\boldsymbol{g}_k(\tilde{\boldsymbol{w}}_k^e(t))\}_{ij}^l \tag{8}$$

$$= -\sum_{j=1}^{n_{\text{in}}^l} \{\boldsymbol{g}_k(\tilde{\boldsymbol{w}}_k^e(t))\}_{ij}^l w_{k,ij}^{e,l}(t), \tag{9}$$

where $Q_{k,i}^{e,l}(t) = \mu_{k,i}^e(t) - \tau_{k,i}^{e,l}(t)$ in (1), (8) is from the definition of pruned parameters in (2) and the unit step function $S(\cdot)$, and (9) is from the identity straight-through estimator [35] to approximate the gradient of the step functions in (8).

From (9), we can see that threshold $\tau_{k,i}^{e,l}$ corresponds to the importance of its connected parameters $w_{k,ij}^{e,l}, 1 \le j \le n_{\text{in}}^l$, in its filter/neuron. This is because the importance of a parameter $w_{ij}^l$ can be estimated by [36]

$$F(\boldsymbol{w}, \boldsymbol{\tau}) - F(\boldsymbol{w}, \tau; w_{ij}^l = 0) \approx g(\boldsymbol{w})_{ij}^l w_{ij}^l, \tag{10}$$

where $F(\boldsymbol{w}, \boldsymbol{\tau}; w_{ij}^l = 0)$ is the loss function when $w_{ij}^l$ is masked and the approximation is obtained from the first Taylor expansion at $w_{ij}^l = 0$. Therefore, if connected parameters were important, the sign of (10) of those parameters will be negative, and the corresponding threshold will decrease as in (9). Otherwise, the threshold will be increased to enforce sparsity. Hence, prematurely pruned parameters will be automatically recovered via a joint optimization of $\boldsymbol{\tau}$ and $\boldsymbol{w}$.

### 3.3.3 Extracting Parameter Importance from Global Thresholds

Since thresholds represent the importance of the connected parameters at each filter/neuron, clients can learn how to prune their parameters from the global thresholds. Moreover, the difference between two consecutive global thresholds $\Delta \boldsymbol{\tau}(t) = \boldsymbol{\tau}(t+1) - \boldsymbol{\tau}(t)$ captures the history of aggregated parameter importance, which can be further used to improve model performance. For instance, from (10), if $\Delta \tau_i^l(t) < 0$, then the parameters connected to threshold $i$ in layer $l$ were globally important. If $\Delta \tau_i^l(t) \ge 0$, then the connected parameters were globally less important. Hence, from $\Delta \boldsymbol{\tau}(t)$, clients can deduce which parameter is globally important or not and further update their model parameters. After generating new global thresholds $\boldsymbol{\tau}(t+1)$, the server broadcasts $\boldsymbol{\tau}(t+1)$ to client $k \in \mathcal{S}_{t+1}$, and then clients calculate $\Delta \boldsymbol{\tau}(t) = \boldsymbol{\tau}(t+1) - \boldsymbol{\tau}(t)$.

We then present how clients can update their model parameters from $\Delta \boldsymbol{\tau}(t)$. For given $\Delta \boldsymbol{\tau}(t)$, we need to decide on the: 1) update direction and 2) update amount. Clients can know the update direction of parameters by considering $\Delta \boldsymbol{\tau}(t)$ and the dominant sign of parameters connected to each threshold. For simplicity, assume that each parameter has a threshold. Then, the gradient of the thresholds in (9) can be rewritten as follows:

$$\boldsymbol{h}_k(\tilde{\boldsymbol{w}}_k(t)) = -\boldsymbol{g}_k(\tilde{\boldsymbol{w}}_k(t))\boldsymbol{w}_k(t). \tag{11}$$

The gradient of the loss $F_k(\tilde{\boldsymbol{w}}_k(t), \boldsymbol{\tau}(t))$ with respect to the whole parameters $\boldsymbol{w}_k(t)$ is given by

$$\frac{\partial F_k(\tilde{\boldsymbol{w}}_k(t), \boldsymbol{\tau}(t))}{\partial \boldsymbol{w}_k(t)} = \boldsymbol{g}_k(\tilde{\boldsymbol{w}}_k(t))|\boldsymbol{w}_k(t)|. \tag{12}$$

From (11) and (12), the gradient direction of a parameter $w$ is opposite of that of its connected threshold if $w > 0$. Otherwise, both the threshold and the parameter have the same gradient direction.

**Algorithm 1:** SpaFL
___
**Input:** Total number of clients $N$; Total communication rounds $T$; Local number of epochs $E$
**Output:** Global thresholds $\boldsymbol{\tau}$ and personalized models $\tilde{\boldsymbol{w}}_k$
**1** The server initializes $\boldsymbol{\tau}(0)$ and $\boldsymbol{w}(0)$ and broadcasts them to every client ;
**2 for** $t = 0$ *to* $T - 1$ **do**
**3** $\quad$ Server randomly samples $\mathcal{S}_t$;
**4** $\quad$ **for** *Client* $k \in \mathcal{S}_t$ **do**
**5** $\quad\quad$ Receive $\boldsymbol{\tau}(t+1)$ from the server and calculate $\Delta\boldsymbol{\tau}(t)$;
**6** $\quad\quad$ Update the current local model using $\Delta\boldsymbol{\tau}(t)$ with (13);
**7** $\quad\quad$ **for** $e = 0$ *to* $E - 1$ **do**
**8** $\quad\quad\quad$ Update $\boldsymbol{w}_k^{e+1}(t) \leftarrow \boldsymbol{w}_k^e(t) - \eta(t)\boldsymbol{g}_k(\tilde{\boldsymbol{w}}_k^e(t)),\ \tilde{\boldsymbol{w}}_k^0(t) = \tilde{\boldsymbol{w}}_k(t)$;
**9** $\quad\quad\quad$ Update $\boldsymbol{\tau}_k^{e+1}(t) \leftarrow \boldsymbol{\tau}_k^e(t) - \eta(t)\boldsymbol{h}_k(\tilde{\boldsymbol{w}}_k^e(t)),\ \boldsymbol{\tau}_k^0(t) = \boldsymbol{\tau}(t)$
**10** $\quad\quad$ Transmit the updated threshold $\boldsymbol{\tau}_k(t)$ to the server
**11** $\quad$ Generate a new global threshold $\boldsymbol{\tau}(t+1)$ using (7)
___

Hence, we can deduce the following: If $w > 0$, the gradient direction of $w$ and the sign of $\Delta\tau$ will have the same sign; otherwise, the gradient direction of $w$ and the sign of $\Delta\tau$ are opposite. In SpaFL, each threshold has multiple connected parameters to its filter/neuron. As such, we decide the update direction of connected parameters by finding the dominant sign among them. To this end, we simply add the connected parameters of each threshold. For instance, consider threshold $i$ in layer $l$ of client $k$, if $\sum_{j=1}^{n_{\text{in}}^l} w_{k,ij}^l(t) > 0$, then the gradient direction of the connected parameters will be the same as the sign of $\Delta\tau_i^l(t)$. Otherwise, it is the opposite of the sign of $\Delta\tau_i^l(t)$. Thus, the update direction can be simply expressed with a XOR operation between the sign of $\Delta\tau_i^l(t)$ and the sign of connected parameters sum. Next, we decide how much a parameter should be updated. From (11) and (12), we can see that a threshold and a parameter have the same magnitude for their gradients. Hence, we simply divide $\Delta\tau_i^l(t)$ by the number of connected parameters $n_{\text{in}}^l$. We finally provide the update equation using $\Delta\boldsymbol{\tau}(t)$ as follows

$$w_{k,ij}^l(t+1) = w_{k,ij}^l(t) + \frac{1}{n_{\text{in}}^l}\Delta\tau_i^l(t)\ \text{XOR}\left\{\text{sign}\left(\sum_{j=1}^{n_{\text{in}}^l} w_{k,ij}^l(t)\right), \text{sign}(\Delta\tau_i^l(t))\right\}, \qquad (13)$$

where $\text{sign}(\cdot)$ is a sign function. This parameter update corresponds to line 7 in Algorithm 1. Note that this additional parameter update is not computationally intensive because it happens only once before local training. We also provide the number of used FLOPs during training with inclusion of this operation in Section 5.

## 4 Theoretical Analysis of SpaFL

We now present our generalization analysis of SpaFL. For the empirical risk $\hat{\mathcal{R}} = \frac{1}{N}\sum_{k=1}^N \frac{1}{D_k}\sum_{i=1}^{D_k} \mathcal{L}(\tilde{\boldsymbol{w}}_k, \boldsymbol{\tau}; z_i)$, we consider the expected risk $\mathcal{R} = \frac{1}{N}\sum_{k=1}^N \mathbb{E}_{z_k\sim\mathcal{D}_k}\mathcal{L}(\tilde{\boldsymbol{w}}_k, \boldsymbol{\tau}; z_k)$, where $\mathcal{L}$ is a loss function and $z$ is an input-output pair. Suppose $\rho_k$ is the ratio of remaining model parameters of client $k$ and $\bar{\rho}$ is the average model density across clients. Then, for the hypothesis $\mathcal{A}(\mathcal{D})$ with global thresholds $\boldsymbol{\tau}$ from Algorithm 1 on the joint training dataset $\mathcal{D} = \cup_{k=1}^N \mathcal{D}_k$ with $\bar{\rho}$, we have the following generalization bound as follows:

**Theorem 1.** *For the loss function* $||\mathcal{L}||_\infty \leq 1$, *the training data size* $D \geq \frac{2}{\epsilon'^2}\ln\left(\frac{16}{\exp(-\epsilon'\delta')}\right)$ *and the total number of communication rounds* $T$, *we have*

$$\mathbb{P}\left[\left|\hat{\mathcal{R}}(\mathcal{A}(\mathcal{D})) - \mathcal{R}(\mathcal{A}(\mathcal{D}))\right| < 9\epsilon'\right] > 1 - \frac{\exp(-\epsilon')\delta'}{\epsilon'}\ln\frac{2}{\epsilon'}, \qquad (14)$$

*where $\epsilon' = \sqrt{2T \log \frac{1}{\delta} \tilde{\epsilon}^2} + T\tilde{\epsilon} \frac{\exp(\tilde{\epsilon})-1}{\exp(\tilde{\epsilon})+1}$,*

$$\delta' = \exp\left(-\frac{\epsilon' + T\tilde{\epsilon}}{2}\right) \left(\frac{1}{1+\exp(\tilde{\epsilon})}\left(\frac{2T\tilde{\epsilon}}{T\tilde{\epsilon}-\epsilon'}\right)\right)^T \left(\frac{T\tilde{\epsilon}+\epsilon'}{T\tilde{\epsilon}-\epsilon'}\right)^{-\frac{\epsilon'+T\tilde{\epsilon}}{2\tilde{\epsilon}}} - \left(1 - \frac{\delta}{1+\exp(\tilde{\epsilon})}\right)^T$$

$$+ 2 - \left(1 - \exp(\tilde{\epsilon})\frac{\delta}{1+\exp(\tilde{\epsilon})}\right)^{\lceil\frac{\epsilon'}{\tilde{\epsilon}}\rceil} \left(1 - \frac{\delta}{1+\exp(\tilde{\epsilon})}\right)^{T-\lceil\frac{\epsilon'}{\tilde{\epsilon}}\rceil}, \tag{15}$$

$$\tilde{\epsilon} = \log\left(\frac{D-\xi}{D} + \frac{\xi}{D}\exp\left(\frac{\sqrt{2}\bar{\rho}M_g\sigma\sqrt{\log\frac{1}{\delta}} + \bar{\rho}^2 M_g^2}{2\sigma^2}\right)\right) \tag{16}$$

*where $\xi$ is the size of a mini-batch, $\sigma$ is the variance of Gaussian noise, and $M_g$ is the maximum diameter of thresholds' gradients* (11). *The proof and the definition of $\delta$ are provided in the Appendix 1.2 and* (12), *respectively.*

From Theorem 1, we can see that, as the average model density $\bar{\rho}$ decreases, the generalization bounds becomes smaller, thereby achieving better generalization performance. This is because $\epsilon'$ and $\tilde{\epsilon}$ decrease as the average model density $\bar{\rho}$ decreases. Hence, SpaFL can improve the generalization performance with sparse models by optimizing and sharing global thresholds.

## 5 Experiments

We now present experimental results to demonstrate the performance, computation costs and communication efficiency of SpaFL. Implementation details are provided in the Supplementary document.

### 5.1 Experiments Configuration

We conduct experiments on three image classification datasets: FMNIST [37], CIFAR-10, and CIFAR-100 [38] datasets with NVIDA A100 GPUs. To distribute datasets in a non-iid fashion, we use Dirichlet (0.2) for FMNIST and Dirichlet (0.1) for CIFAR-10 and CIFAR-100 datasets as done in [39] with $N = 100$ clients. We set the total communication round $T = 500$ and 1500 for FMNIST/CIFAR10 and CIFAR100, respectively. At each round, we randomly sample $K = 10$ clients. Unless stated otherwise, we average all the results over at least 10 different random seeds. We also calculate the best accuracy by averaging each client's performance on its test dataset. For FMNIST dataset, we use the Lenet-5-Caffe. For the Lenet model, we set $\eta(t) = 0.001$, $E = 5$, $\alpha = 0.002$, and a batch size to be 64. For CIFAR-10 dataset, we use a convolutional neural network (CNN) model with seven layers used in [40] with $\eta(t) = 0.01$, $E = 5$, $\alpha = 0.00015$, and a batch size of 16. We adopt the ResNet-18 model for CIFAR-100 dataset with $\eta(t) = 0.01$, $E = 7$, $\alpha = 0.0007$, and a batch size of 64. The learning rate of CIFAR-100 is decayed by 0.993 at each communication round.

### 5.2 Baselines

We compare SpaFL with multiple state of the art baselines that studied sparse model structures in FL. In **FedAvg** [1], every client trains a global dense model and communicates whole model parameters. **FedPM** [28] trains and communicates a binary mask while freezing model parameters. In **HeteroFL** [7], each client trains and communicates $p$-reduced models, which remove the last $1 - p$ output channels in each layer. In **Fjord** [9], each client randomly samples a model from a set of $p$-reduced models, which drops out $p\%$ of filter/neurons in each layer. **FedP3** [41] communicates a subset of sparse layers that are pruned by the server for downlink and personalize the remaining layers. Clients only upload the updated remaining layers to the server. **FedSpa** [4] trains personalized sparse models for clients while maintaining fixed model density during training. **Local** only performs local training with the introduced pruning method without any communications. For the sparse FL baselines, the average target sparsity is set to 0.5 following the configurations in [28, 7, 9, 41, 4].

### 5.3 Main Results

In Table 2 and Fig. 2, we present the averaged accuracy, communication costs, number of FLOPs during training, and convergence rate for each algorithm. We consider all uplink and downlink communications to calculate the communication cost of each algorithm. We also provide the details

| Algorithms | FMNIST | | | CIFAR10 | | | CIFAR100 | | |
|---|---|---|---|---|---|---|---|---|---|
| | Acc | Comm (Gbit) | FLOPs (e+11) | Acc | Comm (Gbit) | FLOPs (e+13) | Acc | Comm (Gbit) | FLOPs (e+14) |
| SpaFL | 89.21±0.25 | **0.1856** | **2.3779** | **69.75±2.81** | **0.4537** | **1.4974** | **40.80±0.54** | **4.6080** | **1.2894** |
| FedAvg | 88.73±0.21 | 133.8 | 10.345 | 61.33±0.15 | 258.36 | 12.382 | 35.51±0.10 | 10712 | 8.7289 |
| FedPM | 63.27± 1.65 | 66.554 | 5.8901 | 52.05± 0.06 | 133.19 | 7.0013 | 28.56 ± 0.15 | 5506.1 | 5.423 |
| HeteroFL | 85.97±0.20 | 68.88 | 5.1621 | 66.83±1.15 | 129.178 | 6.1908 | 37.82±0.15 | 5356.4 | 4.3634 |
| Fjord | 89.08±0.17 | 64.21 | 5.1311 | 66.38±2.01 | 128.638 | 6.1428 | 39.13±0.22 | 5251.4 | 4.1274 |
| FedSpa | **89.30±0.20** | 55.256 | 5.2510 | 67.03±0.63 | 129.31 | 4.2978 | 36.32±0.35 | 5342.2 | 9.275 |
| FedP3 | 89.12±0.14 | 41.327 | 5.8923 | 67.54±0.52 | 67.345 | 6.8625 | 37.73±0.42 | 2682.6 | 4.9384 |
| Local | 84.31±0.20 | 0 | 3.7982 | 57.06±1.30 | 0 | 1.9373 | 33.77±1.87 | 0 | 1.5384 |

Table 2: Performance of SpaFL and other baselines along with their used communication costs (Comm) and computation (FLOPs) resources during whole training.

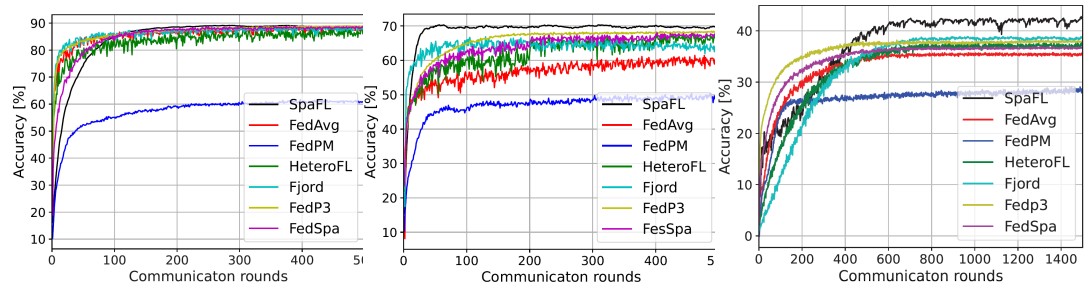

(a) Learning curve on FMNIST    (b) Learning curve on CIFAR-10    (c) Learning curve on CIFAR-100

Figure 2: Learning curves on FMNIST, CIFAR-10, and CIFAR-100

| Algorithm | FMNIST | CIFAR-10 | CIFAR-100 |
|---|---|---|---|
| SpaFL | **89.21±0.25** | **69.75±2.81** | **40.80±0.54** |
| w.o. (13) | 88.20±1.10 | 68.63±1.76 | 38.96±0.80 |

Table 3: Impact of extracting parameter importance from global thresholds

of the FLOPs measure in the Supplementary document. We average the model densities of SpaFL when a model achieved the best accuracy during training. From these results, we observe that SpaFL outperforms all baselines while using the least amount of communication costs and number of FLOPs. The achieved model densities are 35.36%, 30.57%, and 35.38%, for FMNIST, CIFAR-10, and CIFAR-100, respectively. We also observe that SpaFL uses less resources and performs better than FedP3, HetroFL and Fjord, which deployed structured sparse models across clients. For FedP3, clients only upload subset of layers, but the server still needs to send the remaining layers. Although FedPM reduced uplink communication costs by communicating only binary masks, its downlink cost is the same as FedAvg. In SpaFL, since the clients and the server only exchange thresholds, we can significantly reduce the communication costs compared to baselines that exchange the subset of model parameters such as HeteroFL and Fjord. Moreover, SpaFL significantly achieved better performance than Local, which did not communicate trained thresholds. Local achieved 51.2%, 50.1%, and 53.6% model densities for each dataset, respectively. We can see that communicating trained thresholds can make models sparser and achieve better performance. This also corroborates the analysis of Theorem 1. Hence, SpaFL can efficiently improve model performance with small computation and communication costs. In Fig. 2, we show the convergence rate of each algorithm. We can see that the accuracy of SpaFL decreases and then keeps increasing. The initial accuracy drop is from pruning while global thresholds are not trained enough. As thresholds keep being trained and communicated, clients learn how to prune their model, thereby gradually improving the performance with less active filter/neurons.

We provide an empirical comparison between SpaFL and the baseline that does not use the update in Section 3.3.3 in Table. 3. We can see that the update (13) can provide a clear improvement compared to the baseline by extracting parameter importance from global thresholds.

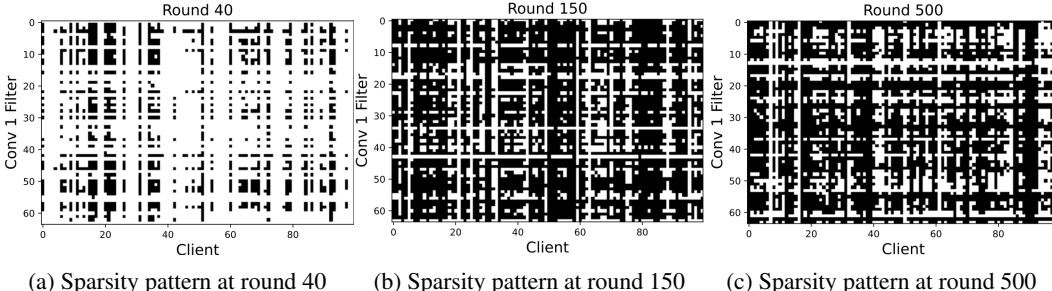

|                                    |                                     |                                     |
|:----------------------------------:|:-----------------------------------:|:-----------------------------------:|
| (a) Sparsity pattern at round 40   | (b) Sparsity pattern at round 150   | (c) Sparsity pattern at round 500   |

Figure 3: Sparsity pattern of conv1 layer on CIFAR-10

| Algorithm | Accuracy [%]      | Density [%]    |
|-----------|-------------------|----------------|
| SpaFL     | **69.78±2.62**    | **42.2±4.8**   |
| FedAvg    | 59.20±0.4         | 100            |

Table 4: Performance of SpaFL with the ViT architecture on CIFAR-10

In Fig. 3, we show the change of structured sparsity of the first convolutional layer with 64 filters with three input channels on CIFAR-10. We color active filters as black and pruned filters as white. We can see that clients learn common sparse structures across training round. For instance, the 31th and 40th filters are all pruned at round 40. Meanwhile, the 20th filter is recovered at rounds 150 and 500. We can know that SpaFL enables clients to learn optimized sparse model structures by optimizing thresholds. In SpaFL, pruned filter/neurons can be recovered by sharing thresholds. At round 40, filters are pruned with high sparsity. Since premature pruning damages the performance, most filters are recovered at round 150. Then, clients gradually enforce more sparsity to filters along with training rounds as shown in Fig. 3c.

In Tab. 4, we show the performance of SpaFL on a vision transformer using the ViT [42] on CIFAR-10 dataset. We used the same data distribution as done in Tab. 2. We apply our pruning scheme to multiheads attention layers. Since a multiheads attention layer essentially consists of stacked linear layers, we can simply use (1), thereby making sparse attention. We can see that SpaFL can be applied to transformer architectures by achieving the density of around 42% while outperforming FedAvg.

## 6 Conclusion

In this paper, we have developed a communication-efficient FL framework SpaFL that allows clients to optimize sparse model structures with low computing costs. We have reduced computational overhead by performing structured pruning through trainable thresholds. To optimize the pruning process, we have communicated only thresholds between clients and a server. We have also presented the parameter update method that can extract parameter importance from global thresholds. Furthermore, we have provided theoretical insights on the generalization performance of SpaFL.

**Limitations and Broader Impact** One limitation of SpaFL is that it cannot explicitly control the sparsity of clients. Since we enforce sparsity through the regularizer term, we need to run multiple experiments to find values for desired sparsity. Another limitation is that our analysis requires a bounded loss function. Meanwhile, in practice, most loss functions may admit bounds that have a large value. For broader impact, SpaFL can reduce not only the computation and communication costs of FL training, but also those of inference phase due to sparsity. Hence, in general, SpaFL can improve the sustainability of FL deployments, and more broadly, of AI.

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

# A  Experiments

## A.1  Implementation Detail

We run all experiments on NVIDIA A100 GPUs with PyTorch. In Table 6, we provide detailed information of model architectures for each dataset. For the FMNIST dataset, we use the Lenet-5-Caffe model, which is Caffe variant of Lenet-5. The Lenet model has 430500 of model parameters and 580 of trainable thresholds. For the CIFAR-10 dataset, we use a CNN model of seven layers used in [40]. It has 807366 of model parameters and 1418 of trainable thresholds. The ResNet-18 model is adopted for the CIFAR-100 dataset with 11159232 of model parameters and 4800 of thresholds. We use a stochastic gradient optimizer with momentum of 0.9. For FMNIST with the Lenet model, we use $\eta(t) = 0.001$, $E = 5$, a batch size of 64, and $\alpha = 0.002$. For CIFAR-10, we use $\eta(t) = 0.01$, $E = 5$, a batch size of 16, and $\alpha = 0.00015$. For CIFAR-100, we use $\eta(t) = 0.01$, $E = 7$ decayed by 0.993 at each communication round, a batch size of 64, and $\alpha = 0.0007$. All trainable thresholds are initialized to zero. We noticed that too large sparsity coefficient $\alpha$ can dominate the training loss, resulting in masking whole parameters in a certain layer. Following the implementation of [31], if a certain layer's density becomes less than 1%, the corresponding trainable thresholds will be reset to zero to avoid masking whole parameters.

For the ViT, we use the patch size of 4, embedding dimension of 128, depth of 6, 8 heads, and set the dimension of linear layers as 256. We use the same setting with the above CIFAR-10 experiments except $\alpha = 0.0001$ and $E = 1$.

|  | FMNIST | CIFAR-10 | CIFAR-100 |
|---|---|---|---|
| **Conv** | (5, 5, out = 20, stride = 1)
Maxpool2d
(5, 5, out = 50, stride = 1)
Maxpool2d | (5, 5, out = 64, stride = 1)
(5, 5, out = 64, stride = 1)
Maxpool2d
(5, 5, out = 128 stride = 1)
(5, 5, out = 128, stride = 1)
Maxpool2d | (3, 3, out = 32, stride = 1)
(3, 3, out = 32, stride = 1) x2
(3, 3, out = 32, stride = 1) x2
(3, 3, out = 64, stride = 2)
(3, 3, out = 64, stride = 1) x3
(3, 3, out = 128, stride = 2)
(3, 3, out = 128, stride = 1) x3 |
| **FC** | (800, 500)
(500, 10) | (512, 128)
(128, 128)
(128, 100) | (256, 100) |

Table 6: Model architectures used in our experiments

### A.1.1  More details about baselines

We compare SpaFL with sparse baselines that investigated structured sparsity. In **FedAvg** [1], every client trains a global dense model and communicates whole model parameters. We used the equal weighted average for the model aggregation. **FedPM** [28] optimizes a binary mask while freezing model parameters. Clients only transmit their arithmetically coded binary masks to the server, and the server broadcasts real-valued probability masks to the clients. We use Adam optimizer with learning rate of 0.1 as done in [28]. **HeteroFL** [7] selects $\lceil pC \rceil$ channels of each layer, where $0 \leq \leq 1$ and $C$ is the number of channels, to make $p$ reduced submodels. Clients train and communicate $p$ reduced submodels during training. We set $p = 0.5$ following [7]. **Fjord** [9] samples $p$ from a uniform distribution $\mathcal{U}(p_{\min}, p_{\max})$. After sampling $p$, clients train $p$ reduced submodel by selecting the first $\lceil pC \rceil$ channels of each layer. We set $p_{\min} = 0.4$ and $\boldsymbol{p}_{\max} = 0.6$ [9]. We provide the learning rates of the baselines in the following table. **FedP3** [41] communicates a subset of sparse layers that are pruned by the server for downlink and personalize the remaining layers. Clients only upload the updated remaining layers to the server. We choose 'OPU2' method, which uniformly selects two layers for clients from the entire network. Hence, clients only upload these chosen layers to the server. For the pruning methods, we adopted the ordered dropout for structured sparsity. **FedSpa** [4] trains personalized sparse models for clients while maintaining fixed model density during training. The initial pruning rate is set to be 0.5 and decayed using cosine annealing.

| Algorithm | FMNIST | CIFAR-10 | CIFAR-100 |
|-----------|--------|----------|-----------|
| FedAvg | $\eta(t) = 0.001$ | $\eta(t) = 0.01$ | $\eta(t) = 0.1$ |
| FedPM | $\eta(t) = 0.15$ | $\eta(t) = 0.1$ | $\eta(t) = 0.1$ |
| HeteroFL | $\eta(t) = 0.001$ | $\eta(t) = 0.005$ | $\eta(t) = 0.01$ |
| Fjord | $\eta(t) = 0.01$ | $\eta(t) = 0.01$ | $\eta(t) = 0.01$ |
| FedP3 | $\eta(t) = 0.01$ | $\eta(t) = 0.01$ | $\eta(t) = 0.01$ |
| FedSpa | $\eta(t) = 0.001$ | $\eta(t) = 0.01$ | $\eta(t) = 0.1$ |
| Local | $\eta(t) = 0.001$ | $\eta(t) = 0.01$ | $\eta(t) = 0.01$ |

Table 7: learning rates used by the baselines

## A.2 Proof of Theorem 1

We next present the detailed proof of Theorem 1. The proof is inspired by [23] and [43] To facilitate the proof, we first provide the definition of differential privacy and key lemmas from [43].

**Definition 1.** *(Differential privacy). A hypothesis $\mathcal{A}$ is $(\epsilon, \delta)$- differentially private for any hypothesis subset $\mathcal{A}_0$ and adjacent datasets $S$ and $S'$ which differ by only one example such that*

$$\log \left[ \frac{\mathbb{P}_{\mathcal{A}(S)}(\mathcal{A}(S) \in \mathcal{A}_0) - \delta}{\mathbb{P}_{\mathcal{A}(S')}(\mathcal{A}(S') \in \mathcal{A}_0)} \right] \leq \epsilon. \tag{17}$$

**Lemma 1.** *(Theorem 4 in [43]) For an iterative algorithm $\mathcal{A}_i$ at round $i$, define the update rule as follows:*

$$\mathcal{M}_i : (\mathcal{A}_{i-1(S),S}) - > \mathcal{A}_i(S). \tag{18}$$

*If for any fixed $\mathcal{A}_{i-1}$, $\mathcal{M}_i$ is $(\epsilon_i, \delta)$ private, then $\{\mathcal{A}_i\}_{i=0}^T$ is $(\epsilon', \delta')$ differntially private such that $\epsilon' = \sqrt{2 \sum_{i=0}^T \epsilon_i^2 \log \frac{1}{\delta}} + \sum_{i=0}^T \epsilon_i \frac{\exp(\epsilon_i)-1}{\exp(\epsilon_i)+1}$,*

$$\delta' = \exp\left(-\frac{\epsilon' + T\epsilon}{2}\right) \left(\frac{1}{1+\exp(\epsilon)} \left(\frac{2T\epsilon}{T\epsilon - \epsilon'}\right)\right)^T \left(\frac{T\epsilon + \epsilon'}{T\epsilon - \epsilon'}\right)^{-\frac{\epsilon' + T\epsilon}{2\epsilon}} - \left(1 - \frac{\delta}{1 + \exp(\epsilon)}\right)^T$$

$$+ 2 - \left(1 - \exp(\epsilon)\frac{\delta}{1 + \exp(\epsilon)}\right)^{\lceil \frac{\epsilon'}{\epsilon} \rceil} \left(1 - \frac{\delta}{1 + \exp(\epsilon)}\right)^{T - \lceil \frac{\epsilon'}{\epsilon} \rceil}, \tag{19}$$

**Lemma 2.** *(Theorem 1 in [43]) For an $(\epsilon, \delta)$ private hypothesis $\mathcal{A}$, the training dataset size $D \leq \frac{2}{\epsilon^2} \ln \frac{16}{\exp(-\epsilon)\delta}$, and the loss function $||\mathcal{L}||_\infty < 1$, we have*

$$\mathbb{P}\left[\left|\hat{\mathcal{R}}(\mathcal{A}(\mathcal{D})) - \mathcal{R}(\mathcal{A}(\mathcal{D}))\right| < 9\epsilon\right] > 1 - \frac{\exp(-\epsilon)\delta}{\epsilon} \ln \frac{2}{\epsilon}, \tag{20}$$

*Proof.* The overall proof follows [23] by showing that SpaFL is an iterative machine learning algorithm that satisfies differential privacy at each round. Then, we can use lemmas from [43] that provide generalization bound to differential private algorithm. One major difference from [23] is that we have global thresholds not global parameters.

We first define notations for the proof. The diameter of the gradient space is defined as $M_g = \max_{w,z,z',\tau} ||\nabla F(w, \boldsymbol{\tau}; z) - \nabla F(w, \boldsymbol{\tau}; z')||$, where $z$ is an input-output pair. We also denote $G_{k,\mathcal{B}} = \frac{1}{|\mathcal{B}|} \sum_{z \in \mathcal{B}} \boldsymbol{h}_k(\tilde{\boldsymbol{w}}_k; z)$ as the average of $\boldsymbol{h}_k(\tilde{\boldsymbol{w}}_k)$ over $\mathcal{B}$. We use $\mathbb{P}$ as probability distribution and $\mathbb{P}^A$ as the probability distribution conditioned on $A$.

From Algorithm 1, it is clear that SpaFL is iteratively optimizing global thresholds $\boldsymbol{\tau}$ in each client at every round. We now derive the differential privacy of (9) in Algorithm 1. Here, each client calculates $\boldsymbol{h}_k$ using its subset of local data. As done in [23], we assume that additive Gaussian noise sample is added in (9) in Algorithm 1 for the analysis. Since we always have global thresholds at round $t$, (9) can be seen as sampling a mini-batch $\mathcal{I}(t)$ from $\mathcal{D} = \cup_k \mathcal{D}$ with mini-batch size $\xi$ and we let

$\mathcal{B}(t) = S_{\mathcal{I}(t)}$ . Then, for fixed $\boldsymbol{\tau}(t-1)$ and two adjacent sample sets $S$ and $S'$, we have

$$\frac{\mathbb{P}^{S_{\mathcal{I}(t)}}(\boldsymbol{\tau}(t) = \boldsymbol{\tau}|\boldsymbol{\tau}(t-1))}{\mathbb{P}^{S'_{\mathcal{I}t}}(\boldsymbol{\tau}(t) = \boldsymbol{\tau}|\boldsymbol{\tau}(t-1))} = \underbrace{\frac{\mathbb{P}^{S_{\mathcal{I}(t)}}(\eta(t-1)G_{S_{\mathcal{I}(t-1)}} + \mathcal{N}(0,\sigma^2\mathbb{I}) = -\boldsymbol{\tau} + \boldsymbol{\tau}(t-1))}{\mathbb{P}^{S'_{\mathcal{I}(t)}}(\eta(t-1)G_{S'_{\mathcal{I}(t-1)}} + \mathcal{N}(0,\sigma^2\mathbb{I}) = -\boldsymbol{\tau} + \boldsymbol{\tau}(t-1))}}_{(A)}, \quad (21)$$

where $\boldsymbol{\tau}(t) = \boldsymbol{\tau}(t-1) - \eta(t-1)\left(G_{S_{\mathcal{I}(t-1)}} + \mathcal{N}(0,\sigma^2\mathbb{I})\right)$ and $G_{S_{\mathcal{I}(t-1)}} = \frac{1}{N}\sum_{k=1}^{N} G_{k,S_{\mathcal{I}_k(t-1)}}$.
We define $\eta(t-1)\boldsymbol{\tau}' = \boldsymbol{\tau}(t-1) - \boldsymbol{\tau}(t) - \eta(t-1)G_{S_{\mathcal{I}(t-1)}}$, then we can rewrite (21) as below

$$(A) = \frac{\mathbb{P}^{S_{\mathcal{I}(t)}}(\mathcal{N}(0,\sigma^2\mathbb{I}) = \boldsymbol{\tau}')}{\mathbb{P}^{S'_{\mathcal{I}(t)}}(G_{S'_{\mathcal{I}(t-1)}} - G_{S_{\mathcal{I}(t-1)}} + \mathcal{N}(0,\sigma^2\mathbb{I}) = \boldsymbol{\tau}')}. \quad (22)$$

Since $\boldsymbol{\tau} \sim \boldsymbol{\tau}(t-1) - \eta(t-1)(G_{S_{\mathcal{I}(t-1)}} + \mathcal{N}(0,\sigma^2\mathbb{I}))$ due to added Gaussian noise samples, $\boldsymbol{\tau}' \sim \mathcal{N}(0,\sigma^2\mathbb{I})$. Then, following the definition of differential privacy, we define

$$D_p(\boldsymbol{\tau}') = \log \frac{\mathbb{P}^{S_{\mathcal{I}(t)}}(\mathcal{N}(0,\sigma^2\mathbb{I}) = \boldsymbol{\tau}')}{\mathbb{P}^{S'_{\mathcal{I}(t)}}(G_{S'_{\mathcal{I}(t-1)}} - G_{S_{\mathcal{I}(t-1)}} + \mathcal{N}(0,\sigma^2\mathbb{I}) = \boldsymbol{\tau}')}$$

$$= -\frac{||\boldsymbol{\tau}'||^2}{2\sigma^2} + \frac{||\boldsymbol{\tau}' - G_{S_{\mathcal{I}(t-1)}} - G_{S'_{\mathcal{I}(t-1)}}||^2}{2\sigma^2} \quad (23)$$

$$= \frac{2\langle\boldsymbol{\tau}', G_{S_{\mathcal{I}(t-1)}} - G_{S'_{\mathcal{I}(t-1)}}\rangle + ||G_{S_{\mathcal{I}(t-1)}} - G_{S'_{\mathcal{I}(t-1)}}||^2}{2\sigma^2}, \quad (24)$$

where (23) is from the definition of Gaussian distribution. We now denote $G_{S_{\mathcal{I}(t-1)}} - G_{S'_{\mathcal{I}(t-1)}}$ in (24) as $\boldsymbol{v}$. We derive the bound of $||\boldsymbol{v}||$ as follows

$$||\boldsymbol{v}|| = ||G_{S_{\mathcal{I}(t-1)}} - G_{S'_{\mathcal{I}(t-1)}}|| = ||\frac{1}{N}\sum_{k=1}^{N} G_{k,S_{\mathcal{I}_k(t-1)}} - G_{k,S'_{\mathcal{I}_k(t-1)}}||$$

$$\leq \frac{1}{N}\sum_{k=1}^{N} ||G_{k,S_{\mathcal{I}_k(t-1)}} - G_{k,S'_{\mathcal{I}_k(t-1)}}||$$

$$\leq \frac{1}{N}\sum_{k=1}^{N} ||\frac{1}{|S_{\mathcal{I}(t-1)}|}\sum_{z\in S_{\mathcal{I}(t-1)}} \boldsymbol{h}_k(\tilde{\boldsymbol{w}}_k(t-1); z) - \frac{1}{|S'_{\mathcal{I}(t-1)}|}\sum_{z\in S'_{\mathcal{I}(t-1)}} \boldsymbol{h}_k(\tilde{\boldsymbol{w}}_k(t-1); z')|| \quad (25)$$

$$\leq \frac{1}{N}\sum_{k=1}^{N} \rho_k M_g = \bar{\rho}M_g, \quad (26)$$

where (25) is from the definition of $G$ and (26) is from the definition of the diameter of gradient $M_g$. Note that some elements of $\boldsymbol{h}_k(\tilde{\boldsymbol{w}}_k; z)$ will be zero since we do not calculate gradients of pruned filter/neurons due to structured sparsity. Hence, we multiply the current model density to derive (26).

We next bound $\langle\boldsymbol{\tau}', \boldsymbol{v}\rangle$ in (24). Since $\langle\boldsymbol{\tau}', \boldsymbol{v}\rangle \sim \mathcal{N}(0, ||\boldsymbol{v}||^2\sigma^2)$, we have the following inequality using Chernoff Bound as

$$\mathbb{P}\left[\langle\boldsymbol{\tau}', \boldsymbol{v}\rangle \geq \sqrt{2}||\boldsymbol{v}|\,\sigma\sqrt{\log 1/\delta}\right] \leq \min_x \exp\left(-\sqrt{2}x||\boldsymbol{v}||\sigma\sqrt{\log 1/\delta}\mathbb{E}[\exp(x\langle\boldsymbol{\tau}', \boldsymbol{v}\rangle)]\right). \quad (27)$$

We define $\delta$ as follows

$$\delta = \min_x \exp\left(-\sqrt{2}x||\boldsymbol{v}||\sigma\sqrt{\log 1/\delta}\mathbb{E}[\exp(x\langle\boldsymbol{\tau}', \boldsymbol{v}\rangle)]\right). \quad (28)$$

Then, with the probability of $1 - \delta$ with respect to $\boldsymbol{\tau}'$, we can derive the bound of (24) as follows

$$D_p(\boldsymbol{\tau}') \leq \frac{\sqrt{2}\bar{\rho}M_g\sigma\sqrt{\log 1/\delta} + \bar{\rho}^2 M_g^2}{2\sigma^2}. \quad (29)$$

Following Lemma 1 and (13) in [23], we can derive that each round in Algorithm 1 is $(\tilde{\epsilon}, \frac{\xi}{D}\delta)$ differntially private, where $\tilde{\epsilon}$ is given as

$$\tilde{\epsilon} = \log\left(\frac{D-\xi}{D} + \frac{\xi}{D}\exp\left(\frac{\sqrt{2}\bar{\rho}M_g\sigma\sqrt{\log\frac{1}{\delta}} + \bar{\rho}^2M_g^2}{2\sigma^2}\right)\right), \tag{30}$$

where $\xi$ is the size of $S_{\mathcal{I}(t-1)}$. Subsequently, we apply Lemma 1 to have $(\epsilon', \delta')$ differential privacy for $T$ communication rounds. Lastly, we finish the proof by using Lemma 2. $\qquad\square$

### A.3 Convergence Rate Analysis

We derive the convergence rate of SpaFL. Since we only communicate thresholds $\boldsymbol{\tau}$, we derive the convergence rate of the global thresholds. In SpaFL, we simultaneously update $\boldsymbol{\tau}$ and $\boldsymbol{w}$, and it is analytically challenging to track the update of $\boldsymbol{\tau}$ for multiple local epochs $E$. As such, we analyze the convergence of SpaFL under the special case with $E = 1$. We leave a more general convergence analysis with multiple local epochs for future works. We now make two assumptions [44] as follows

**Assumption 1.** *(smoothness)* $F_k(\cdot)$ *is $M$-smooth for $\boldsymbol{\tau}$ and client $k$, $\forall k$*

$$F_k(\boldsymbol{w}, \boldsymbol{\tau}') \leq F_k(\boldsymbol{w}, \boldsymbol{\tau}) + \langle \nabla_{\boldsymbol{\tau}}F_k(\boldsymbol{w}, \boldsymbol{\tau}), \boldsymbol{\tau}' - \boldsymbol{\tau}\rangle + \frac{M}{2}||\boldsymbol{\tau}' - \boldsymbol{\tau}||^2, \; \forall\boldsymbol{\tau}. \tag{31}$$

**Assumption 2.** *(Unbiased stochastic gradient) The stochastic gradient $\boldsymbol{h}_k$ is an unbiased estimator of the gradient $\nabla_{\boldsymbol{\tau}}F_k$, respectively, for client $k, \forall k$, such that*

$$\mathbb{E}\boldsymbol{h}_k(\boldsymbol{w}_k) = \nabla_{\boldsymbol{\tau}}F_k(\boldsymbol{w}_k, \boldsymbol{\tau}). \tag{32}$$

Then, we have the following convergence rate

**Theorem 2.** *For $\gamma(t) = \eta(t)(1 - \frac{\alpha(1-M\eta(t))}{2})$ and the largest number of parameters connected to a neuron or filter $n_{in}^{max} > 0$ in a given model, we have*

$$\frac{1}{NT}\sum_{t=0}^{T-1}\sum_{k=1}^{N}\mathbb{E}||\sum\nabla_{\boldsymbol{\tau}}F_k(\tilde{\boldsymbol{w}}_k(t), \boldsymbol{\tau}(t))||^2 \leq \sum_{t=0}^{T-1}\sum_{k=1}^{N}\frac{\mathbb{E}||\nabla_{\boldsymbol{\tau}}F_k(\tilde{\boldsymbol{w}}_k(t), \boldsymbol{\tau}(t)) - \nabla_{\boldsymbol{\tau}_k}F_k(\tilde{\boldsymbol{w}}_k(t), \boldsymbol{\tau}_k(t))||^2}{MNT\gamma(t)}$$

$$+ \sum_{t=0}^{T-1}\frac{2\alpha\eta(t)}{T\gamma(t)}\{1 - M\eta(t)(1 - \alpha)\}||\exp(-\boldsymbol{\tau}(t))||^2$$

$$+ \sum_{t=0}^{T-1}\sum_{k=1}^{N}\frac{M^2\eta(t)^2n_{in}^{max}}{NT\gamma(t)}\mathbb{E}F_k(\tilde{\boldsymbol{w}}_k(t), \boldsymbol{\tau}(t))$$

$$+ \sum_{t=0}^{T-1}\sum_{k=1}^{N}\frac{\mathbb{E}||\boldsymbol{\tau}(t) - \boldsymbol{\tau}_k(t)||^2}{NT\gamma(t)}. \tag{33}$$

From (9), thresholds $\boldsymbol{\tau}(t)$ are updated using parameter gradients $\boldsymbol{g}_k(t), k \in \mathcal{S}$. We can expect that the thresholds will converge when parameters $\boldsymbol{w}_k, \forall k$, converge. We can see that the sparsity regularizer coefficient $\alpha$ impacts convergence. As $\alpha$ increases, we can quickly enforce more sparsity to the model. However, a very large $\alpha$ will damage the performance as $\gamma(t)$ decreases in (1). We can also observed that the convergence depends on the difference between the received global thresholds $\boldsymbol{\tau}(t)$ and the updated thresholds $\boldsymbol{\tau}_k(t)$. Hence, a very large change to the global thresholds will lead to a significantly different binary mask in the next round. Then, local training can be unstable as parameters have to adapt to the new mask. Therefore, from Theorem 2, we can capture the tradeoff between the computing cost and the learning performance in terms of $\alpha$.

*Proof.* We first consider the case in which global thresholds converge. We have the following update rule for global thresholds as

$$\boldsymbol{\tau}(t+1) = \frac{1}{K}\sum_{k\in\mathcal{S}}\boldsymbol{\tau}_k(t) = \boldsymbol{\tau}(t) - \frac{1}{K}\eta(t)\sum_{k\in\mathcal{S}}\boldsymbol{h}_k(\tilde{\boldsymbol{w}}_k(t)) + \alpha\eta(t)\exp(-\boldsymbol{\tau}(t)). \tag{34}$$

We take the expectation over the randomness in client scheduling and stochastic gradients as follows

$$\mathbb{E}\boldsymbol{\tau}(t+1) = \boldsymbol{\tau}(t) - \frac{\eta(t)}{K}\mathbb{E}\sum_{k\in\mathcal{S}}\boldsymbol{h}_k(\tilde{\boldsymbol{w}}_k(t)) + \alpha\eta(t)\exp(-\boldsymbol{\tau}(t)).$$

$$= \boldsymbol{\tau}(t) - \frac{\eta(t)}{N}\mathbb{E}\sum_{k=1}^{N}\nabla_{\boldsymbol{\tau}}F_k(\tilde{\boldsymbol{w}}_k(t),\boldsymbol{\tau}(t)) + \alpha\eta(t)\exp(-\boldsymbol{\tau}(t)). \tag{35}$$

Hence, clearly $\boldsymbol{\tau}$ will eventually converge if $\frac{1}{N}\mathbb{E}||\sum_{k=1}^{N}\nabla_{\boldsymbol{\tau}}F_k(\tilde{\boldsymbol{w}}_k(t),\boldsymbol{\tau}(t))||^2$ converges. We next show that this conditional statement holds in our SpaFL framework.

From the $M$-smoothness of the loss function in Assumption 1, we have

$$F_k(\tilde{\boldsymbol{w}}_k(t),\boldsymbol{\tau}_k(t)) \leq F_k(\tilde{\boldsymbol{w}}_k(t),\boldsymbol{\tau}(t)) + \langle\nabla_{\boldsymbol{\tau}}F_k(\tilde{\boldsymbol{w}}_k(t),\boldsymbol{\tau}(t)),\boldsymbol{\tau}_k(t)-\boldsymbol{\tau}(t)\rangle + \frac{M}{2}||\boldsymbol{\tau}_k(t)-\boldsymbol{\tau}(t)||^2 \tag{36}$$

To facilitate the analysis, we first derive $\boldsymbol{\tau}_k(t) - \boldsymbol{\tau}(t)$ as below

$$\boldsymbol{\tau}_k(t) - \boldsymbol{\tau}(t) = -\eta(t)\boldsymbol{h}_k(\tilde{\boldsymbol{w}}_k(t)) + \alpha\eta(t)\exp(-\boldsymbol{\tau}(t)). \tag{37}$$

Then, we can change (36) as follows

$$\begin{aligned}F_k(\tilde{\boldsymbol{w}}_k(t),\boldsymbol{\tau}_k(t)) \leq{}& F_k(\tilde{\boldsymbol{w}}_k(t),\boldsymbol{\tau}(t)) + \langle\nabla_{\boldsymbol{\tau}}F_k(\tilde{\boldsymbol{w}}_k(t),\boldsymbol{\tau}(t)),-\eta(t)\boldsymbol{h}_k(\tilde{\boldsymbol{w}}_k(t))\rangle\\&+ \langle\nabla_{\boldsymbol{\tau}}F(\tilde{\boldsymbol{w}}_k(t),\boldsymbol{\tau}(t)),\alpha\eta(t)\exp(-\boldsymbol{\tau}(t))\rangle\\&+ \frac{M\eta(t)^2}{2}||\boldsymbol{h}_k(\tilde{\boldsymbol{w}}_k(t)) - \alpha\eta(t)\exp(-\boldsymbol{\tau}(t))||^2.\end{aligned} \tag{38}$$

We next take the expectation to the above inequality and use Assumption 2 as below

$$\begin{aligned}\mathbb{E}F_k(\tilde{\boldsymbol{w}}_k(t),\boldsymbol{\tau}_k(t)) \leq{}& \mathbb{E}F_k(\tilde{\boldsymbol{w}}_k(t),\boldsymbol{\tau}(t)) + \langle\nabla_{\boldsymbol{\tau}}F_k(\tilde{\boldsymbol{w}}_k(t),\boldsymbol{\tau}(t)),-\eta(t)\nabla_{\boldsymbol{\tau}}F_k(\tilde{\boldsymbol{w}}_k(t),\boldsymbol{\tau}(t))\rangle\\&+ \langle\nabla_{\boldsymbol{\tau}}F_k(\tilde{\boldsymbol{w}}_k(t),\boldsymbol{\tau}(t)),\alpha\eta(t)\exp(-\boldsymbol{\tau}(t))\rangle\\&+ \frac{M\eta(t)^2}{2}\mathbb{E}||\boldsymbol{h}_k(\tilde{\boldsymbol{w}}_k(t)) - \alpha\exp(-\boldsymbol{\tau}(t))||^2\\={}& \mathbb{E}F_k(\tilde{\boldsymbol{w}}_k(t),\boldsymbol{\tau}(t)) - \eta(t)||\nabla_{\boldsymbol{\tau}}F_k(\tilde{\boldsymbol{w}}_k(t),\boldsymbol{\tau}(t))||^2\\&+ \underbrace{\alpha\eta(t)(1-M\eta(t))\langle\nabla_{\boldsymbol{\tau}}F_k(\tilde{\boldsymbol{w}}_k(t),\tau(t)),\exp(-\boldsymbol{\tau}(t))\rangle}_{A}\\&+ \underbrace{\frac{M\eta(t)^2}{2}\mathbb{E}||\boldsymbol{h}_k(\tilde{\boldsymbol{w}}_k(t))||^2}_{B} + \frac{M\alpha^2\eta(t)^2}{2}||\exp(-\boldsymbol{\tau}(t))||^2.\end{aligned} \tag{39}$$

We first bound $A$ using $\langle a,b\rangle \leq \frac{||a||^2+||b||^2}{2}$ as below

$$A \leq \frac{\alpha\eta(t)(1-M\eta(t))}{2}\left[||\nabla_{\boldsymbol{\tau}}F_k(\tilde{\boldsymbol{w}}_k(t),\boldsymbol{\tau}(t))||^2 + ||\exp(-\boldsymbol{\tau}(t))||^2\right]. \tag{40}$$

We now further bound $B$ as

$$\begin{aligned}B ={}& \frac{M\eta(t)^2}{2}\mathbb{E}\sum_{l=1}^{L}\sum_{i=1}^{n_{\text{out}}^l}||\sum_{j=1}^{n_{\text{in}}^l}\{\boldsymbol{g}_k(\tilde{\boldsymbol{w}}_k(t))\}_{ij}^l w_{k,ij}^{E-1,l}(t)||^2\\\leq{}& \frac{M\eta(t)^2}{2}\mathbb{E}\sum_{l=1}^{L}\sum_{i=1}^{n_{\text{out}}^l}n_{\text{in}}^l\sum_{j=1}^{n_{\text{in}}^l}||\{\boldsymbol{g}_k(\tilde{\boldsymbol{w}}_k(t))\}_{ij}^l w_{k,ij}^{E-1,l}(t)||^2\\\leq{}& \frac{M\eta(t)^2 n_{\text{in}}^{\max}}{2}\mathbb{E}\sum_{l=1}^{L}\sum_{i=1}^{n_{\text{out}}^l}\sum_{j=1}^{n_{\text{in}}^l}||\{\boldsymbol{g}_k(\tilde{\boldsymbol{w}}_k(t))\}_{ij}^l w_{k,ij}^{E-1,l}(t)||^2\\\overset{(a)}{\leq}{}& \frac{M\eta(t)^2 n_{\text{in}}^{\max}}{2}\mathbb{E}\sum_{l=1}^{L}\sum_{i=1}^{n_{\text{out}}^l}\sum_{j=1}^{n_{\text{in}}^l}||\{\boldsymbol{g}_k(\tilde{\boldsymbol{w}}_k(t))\}_{ij}^l||^2\\={}& \frac{M\eta(t)^2 n_{\text{in}}^{\max}}{2}\mathbb{E}||\boldsymbol{g}_k(\tilde{\boldsymbol{w}}_k(t))||^2 \leq M^2\eta(t)^2 n_{\text{in}}^{\max}F_k(\tilde{\boldsymbol{w}}_k,\boldsymbol{\tau}(t)),\end{aligned} \tag{41}$$

where $n_{\text{in}}^{\max}$ is the largest number of parameters connected to a neuron or filter in a given model, $(a)$ is from $|w| \leq 1$ in Section 3.2.1, and the last inequality is from the $M$-smoothness of $F_k$. By combining $A$ and $B$ with taking expectation, we have

$$\mathbb{E}F_k(\tilde{\boldsymbol{w}}_k(t), \tau_k(t)) \leq \mathbb{E}F_k(\tilde{\boldsymbol{w}}_k(t), \tau(t)) - \eta(t)\left\{1 - \frac{\alpha(1-M\eta(t))}{2}\right\}||\nabla_{\boldsymbol{\tau}}F_k(\tilde{\boldsymbol{w}}_k(t), \boldsymbol{\tau}(t))||^2$$
$$+ \frac{\alpha\eta(t)(1-M\eta(t)(1-\alpha))}{2}||\exp(-\boldsymbol{\tau}(t))||^2 + M^2\eta(t)^2 n_{\text{in}}^{\max}\mathbb{E}F_k(\tilde{\boldsymbol{w}}_k, \boldsymbol{\tau}(t)) \tag{42}$$

By arranging the above inequality, we have

$$||\nabla_{\boldsymbol{\tau}}F(\tilde{\boldsymbol{w}}_k(t), \boldsymbol{\tau}(t))||^2 \leq \frac{1}{\gamma(t)}\left[\mathbb{E}\underbrace{F_k(\tilde{\boldsymbol{w}}_k(t), \tau(t)) - \mathbb{E}F_k(\tilde{\boldsymbol{w}}_k(t), \tau_k(t))}_{(A)}\right]$$
$$+ \frac{\alpha\eta(t)}{\gamma(t)}\left\{1 - M\eta(t)(1-\alpha)\right\}||\exp(-\boldsymbol{\tau}(t))||^2 + \frac{M^2\eta(t)^2 n_{\text{in}}^{\max}}{\gamma(t)}\mathbb{E}F_k(\tilde{\boldsymbol{w}}_k, \boldsymbol{\tau}(t)), \tag{43}$$

where $\gamma(t) = \eta(t)(1 - \frac{\alpha(1-M\eta(t))}{2})$. We now further bound $(A)$ in (43). From Assumption 1, we have the following

$$(A) \leq \langle\nabla_{\boldsymbol{\tau}}F_k(\tilde{\boldsymbol{w}}_k(t), \boldsymbol{\tau}(t)), \boldsymbol{\tau}(t) - \boldsymbol{\tau}_k(t)\rangle + \frac{1}{2M}||\nabla_{\boldsymbol{\tau}}F_k(\tilde{\boldsymbol{w}}_k(t), \boldsymbol{\tau}(t)) - \nabla_{\boldsymbol{\tau}_k}F_k(\tilde{\boldsymbol{w}}_k(t), \boldsymbol{\tau}_k(t))||^2$$
$$\leq \frac{\gamma(t)}{2}||\nabla_{\boldsymbol{\tau}}F_k(\tilde{\boldsymbol{w}}_k(t), \boldsymbol{\tau}(t))||^2 + \frac{1}{2\gamma(t)}||\boldsymbol{\tau}(t) - \boldsymbol{\tau}_k(t)||^2$$
$$+ \frac{1}{2M}||\nabla_{\boldsymbol{\tau}}F_k(\tilde{\boldsymbol{w}}_k(t), \boldsymbol{\tau}(t)) - \nabla_{\boldsymbol{\tau}_k}F_k(\tilde{\boldsymbol{w}}_k(t), \boldsymbol{\tau}_k(t))||^2. \tag{44}$$

Based on (44), we can bound (43) as below

$$||\nabla_{\boldsymbol{\tau}}F(\tilde{\boldsymbol{w}}_k(t), \boldsymbol{\tau}(t))||^2 \leq \frac{1}{M\gamma(t)}\mathbb{E}||F_k(\nabla_{\boldsymbol{\tau}(t)}\tilde{\boldsymbol{w}}_k(t), \tau(t)) - \nabla_{\boldsymbol{\tau}_k(t)}F_k(\tilde{\boldsymbol{w}}_k(t), \tau_k(t))||^2$$
$$+ \frac{2\alpha\eta(t)}{\gamma(t)}\left\{1 - M\eta(t)(1-\alpha)\right\}||\exp(-\boldsymbol{\tau}(t))||^2 + \frac{2M^2\eta(t)^2 n_{\text{in}}^{\max}}{\gamma(t)}\mathbb{E}F_k(\tilde{\boldsymbol{w}}_k, \boldsymbol{\tau}(t))$$
$$+ \frac{||\boldsymbol{\tau}(t) - \boldsymbol{\tau}_k(t)||^2}{\gamma(t)^2}. \tag{45}$$

From (45), we can bound the averaged aggregated gradients with respect to thresholds as below

$$\frac{1}{N}\mathbb{E}||\sum_{k=1}^{N}\nabla_{\boldsymbol{\tau}}F(\tilde{\boldsymbol{w}}_k(t), \boldsymbol{\tau}(t))||^2 \leq \frac{1}{N}\sum_{k=1}^{N}\mathbb{E}||\nabla_{\boldsymbol{\tau}}F(\tilde{\boldsymbol{w}}_k(t), \boldsymbol{\tau}(t))||^2$$
$$\leq \frac{1}{NM\gamma(t)}\left(\sum_{k=1}^{N}\mathbb{E}||F_k(\nabla_{\boldsymbol{\tau}(t)}\tilde{\boldsymbol{w}}_k(t), \tau(t)) - \nabla_{\boldsymbol{\tau}_k(t)}F_k(\tilde{\boldsymbol{w}}_k(t), \tau_k(t))||^2\right)$$
$$+ \frac{2\alpha\eta(t)}{\gamma(t)}\left\{1 - M\eta(t)(1-\alpha)\right\}||\exp(-\boldsymbol{\tau}(t))||^2$$
$$+ \frac{1}{N}\sum_{k=1}^{N}\frac{2M^2\eta(t)^2 n_{\text{in}}^{\max}}{\gamma(t)}\mathbb{E}F_k(\tilde{\boldsymbol{w}}_k, \boldsymbol{\tau}(t)) + \frac{1}{N}\sum_{k=1}^{N}\frac{\mathbb{E}||\boldsymbol{\tau}(t) - \boldsymbol{\tau}_k(t)||^2}{\gamma(t)^2}. \tag{46}$$

By summing the above inequality from $t = 0$ to $T - 1$, we can obtain the result of Theorem 2. $\quad\square$

Based on Theorem 2, we can derive the convergence rate with the Big-O notation by following the steps in [8]. We first assume $\eta = 1/\sqrt{T}$ and $0 \leq \alpha < 1$. Then, we can bound $\frac{1}{\gamma(t)} \leq \frac{2}{\eta(t)(1-\alpha)}$. By

replacing $\eta(t)$ and $\gamma(t)$ with their assumed value and bound into the above convergence rate, we have the following bound

$$\mathcal{O}(\frac{A}{\sqrt{T}(1-\alpha)}) + \mathcal{O}(\frac{B}{T(1-\alpha)}) + \mathcal{O}(\frac{C}{\sqrt{T}}) + \mathcal{O}(\frac{D}{\sqrt{T}}), \tag{47}$$

where $A = \sum_{t=0}^{T-1} \sum_{k=1}^{N} \frac{\mathbb{E}||\nabla_\tau F_k(\tilde{w}_k(t),\tau(t)) - \nabla_{\tau_k} F_k(\tilde{w}_k(t),\tau(t))||^2}{MN}$, $B = \sum_{t=0}^{T-1} 4\alpha|| \exp(-\tau(t))||^2$, $C = M^2 n_{in}^{max} G^2/2$, and $D = \sum_{t=0}^{T-1} \sum_{k=1}^{N} \frac{||\tau(t)-\tau_k(t)||^2}{N}$.

### A.4 Communication Costs Measure

We calculate the communication cost of SpaFL considering both uplink and downlink communications. At each round $t$, sampled clients transmit their updated thresholds to the server. Hence, the uplink communication costs can be given by

$$\text{Comm}_{\text{Up}} = K \times \boldsymbol{\tau}_{\text{num}} \times 32 \text{ [bits]}, \tag{48}$$

where $\boldsymbol{\tau}_{\text{num}}$ is the number of thresholds of a given model. In downlink, the server broadcasts the updated global threshold to sampled clients. Hence, the downlink communication costs can be given as below

$$\text{Comm}_{\text{down}} = K \times \boldsymbol{\tau}_{\text{num}} \times 32 \text{ [bits]}. \tag{49}$$

Therefore, total communication costs can be given by $T \times (\text{Comm}_{\text{Up}} + \text{Comm}_{\text{down}})$.

### A.5 FLOPs Measure

We calculate the number of FLOPs during training using the framework introduced in [32]. We consider a convolutional layer with an input tensor $X \in \mathbb{R}^{N \times C \times X \times Y}$, parameter tensor $W \in \mathbb{R}^{F \times C \times R \times S}$, and output tensor $O \in \mathbb{R}^{N \times F \times H \times W}$. Here, the input tensor $X$ consists of $N$ number of samples, each of which has $X \times Y$ dimension. The parameter tensor $W$ has $F$ filters of $C$ channels with kernel size $R \times S$. The output tensor $O$ will have $F$ output channels with dimension $H \times W$ for $N$ samples. During forward propagation, a filter in $W$ performs convolution operation with the input tensor $X$ to produce a single value in the output tensor $O$. Hence, we can approximate the number of FLOPs as $N \times (C \times R \times S) \times F \times H \times W$. Since we use a sparse model during forward propagation, the number of FLOPs can be reduced to $\rho \times N \times (C \times R \times S) \times F \times H \times W$, where $\rho = \frac{||\boldsymbol{p}||_0}{||W||_0}$ is the density of the parameter matrix $W$. For the backpropagation, we calculate it as 2 times of that of forward propagation following [45].

For a fully connected layer with input tensor $X \in \mathbb{R}^{N \times X}$ and parameter tensor $W \in \mathbb{R}^{X \times Y}$, the input tensor $X$ is multiplied with $W$ during the forward propagation. Hence, with the density of $W$, we can calculate the number of FLOPs for the forward propagation as $\rho \times N \times X \times Y$. In backpropagation, we follow the same process for convolutional layers.

We also consider the number of FLOPs to perform line 6 in Algorithm 1 for updating the local models from global thresholds. Sampled clients first have to decide update directions by doing summation of connected parameters at each neuron/filter (sum operation). Then, they update their local models using the received global thresholds (sum and multiply operations). This corresponds to $1.5 \times d$ FLOPs, where $d$ is the number of model parameters. Then, the total number of FLOPs during one local epoch at round $t$ can be approximately given by

$$\text{FLOP}(t) = \sum_{l=1}^{L} 3N \times (C_l \times R_l \times S_l) \times F_l \times H_l \times W_l \times \mathbb{1}\{\text{layer } l == \text{conv}\}$$
$$+ 3 \times N \times X_l \times Y_l \times \mathbb{1}\{\text{layer } l == \text{fc}\} + 1.5d \tag{50}$$

## B Change of sparsity patterns on CIFAR-10

Here, we present the change of sparsity patterns of different layers on CIFAR-10.



Figure 4: Sparsity patterns of conv2 layer on CIAFR-10



Figure 5: Sparsity patterns of dense1 layer on CIAFR-10

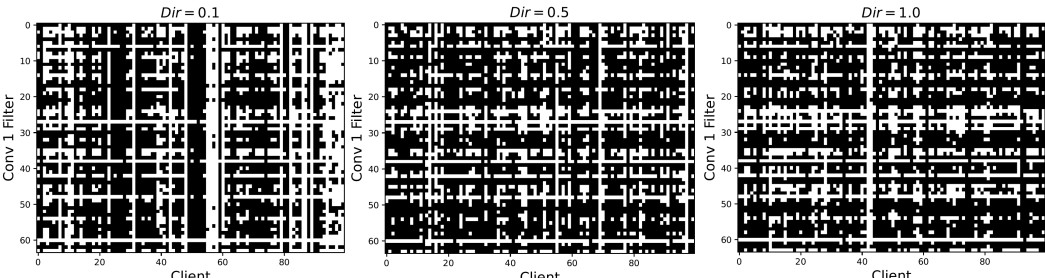

Figure 6: Sparsity patterns of conv1 layer on CIAFR-10

### B.1    Change of Model Sparsity patterns in conv2

### B.2    Change of Model Sparsity patterns in dense1

From Figs. 4 and 5, we can observe that clients learn common sparsity patterns across layers by communicating thresholds.

### B.3    Change of Model Sparsity patterns with different data heterogeneity

From Fig. 6, we can see that as the data heterogenity decreases, clients share more similar sparsity patterns across their filters.

