# OpenReview forum: "SpaFL: Communication-Efficient Federated Learning With Sparse Models And Low Computational Overhead"
_NeurIPS.cc/2024/Conference — NeurIPS 2024 poster_

### Official Review · Reviewer_Ju3Y · 2024-06-13

**Soundness:** 3
**Presentation:** 3
**Contribution:** 3
**Rating:** 6
**Confidence:** 3

**Summary:**

The authors consider distributed training of sparse models, by optimizing over the thresholds used to prune the models.

**Strengths:**

I am not an expert of deep learning but the results look convincing enough.

**Weaknesses:**

I would be interested in a discussion and comparison with other approaches that train sparse models, such as
* Meinhardt et al. "Prune at the Clients, Not the Server: Accelerated Sparse Training in Federated Learning," preprint arXiv:2405.20623, 2024.
* Yi et al. "FedComLoc: Communication-Efficient Distributed Training of Sparse and Quantized Models," preprint arXiv:2403.09904, 2024.

and references therein.

The experiments should investigate more the influence of the level of heterogeneity, because this is what makes it difficult to identify the sparsity pattern.

**Questions:**

No question

**Limitations:**

The technical limitations are discussed at the end of the paper.

---

> ### Author Rebuttal · Authors · 2024-08-07
>
> Thank you for the constructive comments. We provide our response for each question below.
>
> >Q1:I would be interested in a discussion and comparison with other approaches that train sparse models, such as
>
> Meinhardt et al. "Prune at the Clients, Not the Server: Accelerated Sparse Training in Federated Learning," preprint arXiv:2405.20623, 2024.
> Yi et al. "FedComLoc: Communication-Efficient Distributed Training of Sparse and Quantized Models," preprint arXiv:2403.09904, 2024.
> and references therein.
>
> **A1:** We compare SpaFL with two more baselines: FedP3 [1] and FedSpa [2]. Specifically, FedP3 communicates a subset of sparse layers that are pruned by the server and personalize the remaining layers. FedSpa trains a personalized sparse models while fixing the model density during the training, For FedP3, we set global pruning ratio as 0.5 and use OPU2 method (overlaps two layers) as done in [5]. We use the same data distribution and model architecture as done in our experiment, For CIFAR-10, we set the learning rate as 0.01. For CIFAR-100, we set the learning rate as 0.1 and decayed it by 0.997 for each communication round, We set the initial pruning ratio of FedSpa as 0.5 with cosine annealing as done in [6]. We set the learning rate as 0.01 and 0.1 for CIFAR-10 and CIFAR-100, respectively.
>
> |Algorithm | Accuracy (CIFAR-10) | Accuracy (CIFAR-100)|
> | --- | --- | ---|
> | SpaFL  | $\mathbf{69.75 \pm 2.81}$ |$\mathbf{40.80 \pm 0.54}$|
> |FedP3 | $67.54 \pm 0.52$ | $37.73 \pm 0.42$|
> | FedSpa   | $67.03 \pm 0.63$ | $36.32 \pm 0.35$|
>
> >Q2: The experiments should investigate more the influence of the level of heterogeneity, because this is what makes it difficult to identify the sparsity pattern.
>
> **A2:** As the data distribution becomes iid, the sparsity pattern will be more similar. Due to the limited time frame, we will update the result shortly.
>
> **References**
>
> [1] Yi, Kai, et al. "FedP3: Federated Personalized and Privacy-friendly Network Pruning under Model Heterogeneity." The Twelfth International Conference on Learning Representations, 2024
>
> [2] Huang, Tiansheng, et al. "Achieving personalized federated learning with sparse local models." arXiv preprint arXiv:2201.11380 (2022).

---

> > ### Comment · Reviewer_Ju3Y · 2024-08-08
> >
> > I have read the rebuttal. I think the paper contains interesting insights on the difficult problem of sparse training so I am keeping my score.

---

> > > ### Author Response · Authors · 2024-08-14
> > >
> > > We appreciate the reviewer for the constructive comment. We will update the revised version with newly added baselines and sparsity patterns with decreasing data heterogeneity.

---

### Official Review · Reviewer_qu5D · 2024-07-08

**Soundness:** 2
**Presentation:** 3
**Contribution:** 3
**Rating:** 6
**Confidence:** 3

**Summary:**

This paper introduces SpaFL, a federated learning framework that enhances communication efficiency and minimizes computational overhead by optimizing sparse model structures. They achieve this goal by defining a trainable threshold which leads to structured sparsity. Since the server and clients only exchange thresholds, SpaFL is able to reduce communication overhead. Furthermore, these trainable thresholds also lead to low computing costs.

**Strengths:**

1. The paper is generally well-written and easy to follow

2. The idea of using trainable thresholds is new and effective for achieving both communication efficiency and low computational overhead.

3. They perform valid experiments with the baseline methods and some image datasets. The accuracy performance of their method is better than others.

**Weaknesses:**

1. I recommend the authors to do additional experiments with NLP tasks.

2. In Figure 8 of the FedPM paper, the accuracy achieved by FedPM is higher than what is reported in this paper. Could you clarify the reason for this discrepancy? Is it due to differences in experimental setup and hyperparameters? If so, could you ensure that all conditions are consistent for the CIFAR-100 experiments and compare your method to the original accuracy reported in the FedPM paper?

**Questions:**

1. If you apply this method to the transformer based models, how can you define trainable thresholds? To be specific, you gave an explanation for a convolutional layer in section 3.1, and can we just similarly define for self attention modules?

---

> ### Author Rebuttal · Authors · 2024-08-07
>
> Thank you for the constructive comments.  We provide our response to each comment below.
>
> >Q1: In Figure 8 of the FedPM paper, the accuracy achieved by FedPM is higher than what is reported in this paper. Could you clarify the reason for this discrepancy? Is it due to differences in experimental setup and hyperparameters? If so, could you ensure that all conditions are consistent for the CIFAR-100 experiments and compare your method to the original accuracy reported in the FedPM paper?
>
> **A1**: In Figure 8 of the FedPM paper, the authors distributed IID CIFAR-100 dataset over 10 clients, and every clients are sampled at each communication round, Meanwhile, in our setting, we distributed non-iid CIFAR-100 dataset with Dirichlet distribution of 0.1 over 100 clients and only sampled 10 clients at each communication round. We would like to note that every algorithm has different well working configurations. In the implementation of the FedPM, the authors used Adam optimizer. Since we were not able to reproduce their results with SGD optimizer, we used the Adam optimizer for the FedPM in Table 2. Next, we compare the result of SpaFL over IID CIFAR-100 dataset over 10 clients with the reported accuracy in the FedPM paper. We sampled every client at each communication round and set the learning rate as 0.05 and decayed it by 0.993 at each round. We set local epoch as 7 and averaged over 3 runs.
>
> | Algorithm |Accuracy |
> | --- | --- |
> | SpaFL | $\mathbf{45.32 \pm 0.3}$ |
> | FedPM | 42 |
>
> >Q2:I recommend the authors to do additional experiments with NLP tasks.
>
> **A2:** Due to the limited time frame, we will update the result with a transformer model shortly.
>
> >Q3: If you apply this method to the transformer based models, how can you define trainable thresholds? To be specific, you gave an explanation for a convolutional layer in section 3.1, and can we just similarly define for self attention modules?
>
> **A3:** A transformer model mostly consists of multi-head attention modules and feed-forward network modules. In SpaFL, we already provided pruning methods for feed-forward networks such as Linear and Convolutional layers. We can use a similar pruning method for multi-head attention modules as we did for Linear layers. Specifically, we defined a trainable threshold for each column of a matrix in a linear layer, We pruned a whole column if the average magnitude is smaller than its threshold, thereby achieving structured sparsity. Since multi-head attention modules also consist of multiple matrices, we can use the same pruning method. For each column of query, key, value and projection matrices, we can define a trainable threshold, If the average magnitude of parameters in a column is smaller than its threshold, we can prune the entire column. Due to the limited time frame, we will update the result with NLP tasks shortly,

---

> > ### Author Response · Authors · 2024-08-11
> >
> > As the reviewer mentioned, the current algorithm can be applied to transformer models. Since multi-head attention modules consist of multiple linear matrices: query, key, and value, we can use the same pruning approach as we did with linear layers in SpaFL. For each column of query, key, value matrices, we can define a trainable threshold, If the average magnitude of parameters in a column is smaller than its threshold, we can prune the entire column. We tested this with a vision transformer with depth of 6, heads of 8, dim of 128, and linear layer dimension of 256 on non-iid CIFAR-10 dataset over 100 clients with dirichelet distribution of 0.1. We briefly used the same hyperparameters as we did in CIFAR-10 experiment in our manuscript without hyperparameter optimization and averaged the results over three random seeds.
> >
> > | Algorithm   | Accuracy (CIFAR-10) | Density|
> > | -------- | ------- | --------|
> > | SpaFL  | $68.6 \pm 1.5$    | 54.7%|
> > | FedAvg | $59.2 \pm 0.4$     | 100%|
> >
> > Hence, the proposed algorithm can be applied to transformer models.

---

> > > ### Comment · Reviewer_qu5D · 2024-08-11
> > > **raise to 6**
> > >
> > > Thank you for your rebuttal and additional experiments. I raised my score to 6 since my questions are well addressed.

---

> > > > ### Author Response · Authors · 2024-08-14
> > > >
> > > > We appreciate the reviewer for the constructive comment. We will update the revised version with the proposed method for transformer models.

---

### Official Review · Reviewer_28Jj · 2024-07-12

**Soundness:** 2
**Presentation:** 1
**Contribution:** 2
**Rating:** 5
**Confidence:** 4

**Summary:**

This paper suggests communicating the threshold instead of the model parameters in federated learning. Through empirical validations on popular benchmarks, the proposed method, SpaFL, is shown to have lower computational overhead and achieve relatively good results.

**Strengths:**

Communicating the threshold instead of the model weights seems to be interesting. Preliminary empirical results on popular benchmarks seem to have shown that this idea works with lower communication overhead and relatively good performance.

**Weaknesses:**

1. The main figure, Figure 1, is not informative and very confusing. I would expect to see a high-level explanation of how the threshold for each client is selected, whether they are the same for each layer, and how the selected thresholds are combined. At this moment, I can only see that all local models have the same threshold and they are directly combined at the server. Due to the misalignment between the main figure and the illustration in section 3.1, the necessary details to fully understand the main idea of this paper are not easy to follow. The writing in this section could also be improved.

2. In Theorem 1, the authors provided a generalization bound. However, the intuition and analysis for this bound are missing. At this moment, I do not have a good sense of, in a practical or special case, how large the right-hand side probability of Equation 14 is? How large must the minimal training data size $D$ be? Is it practical? Also, this bound only measures the distance between the empirical risk and the expected risk; how can this be regarded as a generalization bound? To my understanding, a generalization bound should be between the training set and the testing set. Furthermore, could you please provide the convergence rate and communication complexity of the proposed method? These are very important for the theoretical analysis of FL.

3. The baselines are outdated, such as Fjord (NeurIPS21) and HeteroFL (ICLR21). Can you please compare more recent methods, which can be easily found, such as [FedPAC](https://arxiv.org/abs/2306.11867) and [FedCR](https://proceedings.mlr.press/v202/zhang23w.html)? I would expect to see more comparisons with newer methods. They do not need to be the ones mentioned above, but I believe it is critical to determine whether the proposed method is still interesting to explore at this moment.

**Questions:**

I would expect a README file to better understand your code efficiently.

**Limitations:**

The major limitations include the unclear presentation of the proposed method, lack of analysis of the generalization bound, absence of convergence and communication efficiency analysis, and the use of outdated empirical baselines.

---

> ### Author Rebuttal · Authors · 2024-08-07
>
> Thank you for the constructive comments. We provide our response to each comment below.
> >Q1: I would expect to see a high-level explanation of how the threshold for each client is selected, whether they are the same for each layer, and how the selected thresholds are combined.
>
> **A1:**  In SpaFL, all thresholds are initialized to zero. Each threshold will be updated by (6) during the training. As such, each threshold will have different value. We averaged selected thresholds to generate global thresholds for the next round as shown in (7). We will clarify Figure. 1 and Section 3.1.
>
> >Q2: This bound only measures the distance between the empirical risk and the expected risk; how can this be regarded as a generalization bound? To my understanding, a generalization bound should be between the training set and the testing set.
>
> **A2:** It is our understanding that the generalization bound is the difference between empirical and expected risks as shown in [1, 2].
>
> >Q3:  Is it practical?  how large the right-hand side probability of Equation 14 is? How large must the minimal training data size D be?
>
> **A3**: Theorem 1 can provide a probabilistic guarantee with a certain amounts of data and communication rounds under the assumption of the bounded loss function, As shown in the result and the proof, the result depends on theoretical values such as $(\epsilon, \delta)$ differential privacy, gaussian noise $\sigma$ and the maximum diameter of gradient space $M_g$, To calculate the right-hand side probability of (14), we first assume $\epsilon =4, \delta = 1/10^6$, $\sigma = 2$, and $M\_g =1$ from [1, 2]. We also assume mini-batch size $\xi=64$ and do $T=500$ communication rounds with $\rho = 0.5$ average model density. Then, we can obtain the right-hand side probability as 0.61 for bound the generalization error around as 0.25, and the minimum required training sample $D$ is around 10000.
>
> >Q4: Furthermore, could you please provide the convergence rate and communication complexity of the proposed method?
>
> **A4**: We provided the training curve in Figure 2. We can see that the proposed approach shows relatively faster convergence rate compared to the baselines,
>
> >Q5: The baselines are outdated, such as Fjord (NeurIPS21) and HeteroFL (ICLR21). Can you please compare more recent methods, which can be easily found, such as FedPAC and FedCR? I would expect to see more comparisons with newer methods. They do not need to be the ones mentioned above, but I believe it is critical to determine whether the proposed method is still interesting to explore at this moment.
>
> **A5**: We compare SpaFL with two more baselines: FedP3 [5] and FedSpa [6]. Specifically, FedP3 communicates a subset of sparse layers that are pruned by the server and personalize the remaining layers. FedSpa trains a personalized sparse models while fixing the model density during the training, For FedP3, we set global pruning ratio as 0.5 and use OPU2 method (overlaps two layers) as done in [5]. We use the same data distribution and model architecture as done in our experiment, For CIFAR-10, we set the learning rate as 0.01. For CIFAR-100, we set the learning rate as 0.1 and decayed it by 0.997 for each communication round, We set the initial pruning ratio of FedSpa as 0.5 with cosine annealing as done in [6]. We set the learning rate as 0.01 and 0.1 for CIFAR-10 and CIFAR-100, respectively.
>
> |Algorithm | Accuracy (CIFAR-10) | Accuracy (CIFAR-100)|
> | --- | --- | ---|
> | SpaFL  | $\mathbf{69.75 \pm 2.81}$ |$\mathbf{40.80 \pm 0.54}$|
> |FedP3 | $67.54 \pm 0.52$ | $37.73 \pm 0.42$|
> | FedSpa   | $67.03 \pm 0.63$ | $36.32 \pm 0.35$|
>
>
>
> **References**
>
> [1] Dupuis, Benjamin, George Deligiannidis, and Umut Simsekli. "Generalization bounds using data-dependent fractal dimensions." International Conference on Machine Learning, 2023.
>
> [2] Chu, Yifeng, and Maxim Raginsky. "A unified framework for information-theoretic generalization bounds." Advances in Neural Information Processing Systems 36 (2023): 79260-79278.
>
> [3] Abadi, Martin, et al. "Deep learning with differential privacy." Proceedings of the 2016 ACM SIGSAC conference on computer and communications security. 2016.
>
> [4] Balle, Borja, Gilles Barthe, and Marco Gaboardi. "Privacy amplification by subsampling: Tight analyses via couplings and divergences." Advances in neural information processing systems 31 (2018).
>
> [5] Yi, Kai, et al. "FedP3: Federated Personalized and Privacy-friendly Network Pruning under Model Heterogeneity." The Twelfth International Conference on Learning Representations, 2024
>
> [6] Huang, Tiansheng, et al. "Achieving personalized federated learning with sparse local models." arXiv preprint arXiv:2201.11380 (2022).

---

> > ### Comment · Reviewer_28Jj · 2024-08-12
> >
> > Thank you for your rebuttal and for clarifying the threshold settings and the generalization bound in your paper. However, I still have concerns regarding the theoretical analysis and comparison with baseline methods.
> >
> > **Convergence Rate and Communication Complexity**: My concern pertains to the lack of theoretical analysis concerning the convergence rate and communication complexity of your proposed method. Currently, your paper and theorem do not include an analysis or theoretical comparison with existing works. I am looking for a more comprehensive analysis and a detailed theoretical comparison with established studies.
> >
> > **Baseline Comparison**: While I appreciate the inclusion of baselines such as FedSpa (2022) and FedP3 (ICLR24) in your rebuttal, the main comparison with Fjord (NeurIPS21) and HeteroFL (ICLR21) in your paper and FedSpa (2022) still seems outdated for a NeurIPS24 submission. Regarding FedP3, I understand the rationale behind selecting a global pruning rate of 0.5. However, I must point out a discrepancy. FedP3 employs personalized model aggregation not for enhancing performance but for significantly reducing communication costs and maintaining privacy. Comparing this with your method, which uses full-layer aggregation, to FedP3’s partial-aggregation is *potentially unfair and misleading*. It is essential to clarify this point. Since the FedP3 study demonstrated that reducing the global pruning rate from 0.9 to 0.5 significantly impacted performance, the results you presented are not surprising. I would like to see a comparison that includes trends in communication costs between FedP3 and your method. Additionally, what would the performance of FedP3 look like with a higher global pruning rate? Alternatively, as suggested in my initial review, consider comparing your method with more recent baselines to clearly demonstrate the effectiveness of your method.

---

> > > ### Author Response · Authors · 2024-08-13
> > >
> > > Thank you for the constructive feedbacks. We first provide the convergence analysis. For the derivation, we need few conventional theoretical assumptions following [1] [2].
> > >
> > > **A1**: Loss function $F_k(\cdot)$ is $M$ smooth for $\tau$ and client $k, \forall k$.
> > >
> > > **A2**: The stochastic gradient $h_k$ is an unbiased estimator of the gradient $\nabla_{\tau} F_k$ for client $k, \forall k$ such that $\mathbb{E} h_k(w_k) = \nabla_{\tau} F_k(w_k, \tau)$
> > >
> > > **A3**: There exists $G \geq 0$ such that $\mathbb{E}||g_k(w_k)||^2 \leq G^2, \forall k$.
> > >
> > > In SpaFL, both parameters and thresholds are simultaneously updated, but only thresholds are communicated between clients and the server. Following the same notations in the paper, we assume $N$ number of clients , learning rate $\eta$, $T$ number of communication rounds, sparsity regularizer coefficient $\alpha$. and the thresholds update rule $\tau\_k(t) \leftarrow \tau(t) - \eta(t) h\_k(\tilde{w}\_k(t)) + \alpha \exp(-\tau(t))$. Then, we have the following convergence rate.
> > >
> > > For $\gamma(t) = \eta(t) (1 - \frac{\alpha (1 - M \eta(t) }{2} )$ and the largest number of parameters connected to a filter/neuron $n_{in}^{max} >0$ in a given model, we have
> > >
> > > $\frac{1}{NT} \sum_{t=0}^{T-1} \mathbb{E} || \sum_{k=1}^N \nabla_{\tau} F_k (\tilde{w}_k(t), \tau(t)) ||^2 $
> > >
> > > $\leq \sum_{t=0}^{T-1}  \sum_{k=1}^N \frac{   \mathbb{E} ||  \nabla_{\tau} F_k (\tilde{w}\_k(t), \tau(t) ) - \nabla_{\tau_k} F_k ( \tilde{w}_k(t), \tau(t) ) ||^2} {  {MNT\gamma(t)}}$
> > > $+  \sum\_{t=0}^{T-1}  \frac{ 2\alpha \eta(t) }{T \gamma(t) } (1 - M \eta(t)(1 - \alpha) ) || \exp(-\tau(t) ) ||^2$
> > >
> > > $ + \sum\_{t=0}^{T-1} \frac{M^2 \eta(t)^2n\_{in}^{max} }{2 T \gamma(t)} G^2 $ + $\sum_{t=0}^{T-1}  \sum_{k=1}^N \frac{\mathbb ||\tau(t) - \tau\_k(t)||^2 }{NT\gamma(t)}$.
> > >
> > > In SpaFL, we enforce sparsity through the sparsity regularizer $\alpha$. We can see that as $\alpha$ increases, the convergence rate can be damaged due to the second and the last terms on the right-hand side. Hence, a large $\alpha$ can induce more sparsity to models, but it can damage the performance. We can also see that the convergence rate depends on the difference of gradients between global and local thresholds as shown in the first term, thereby capturing the impact of the non-iid dataset. We will provide the detailed proof in the appendix of the revised version,
> > >
> > > [1] Li, Xiang, et al. "On the convergence of fedavg on non-iid data." arXiv preprint arXiv:1907.02189 (2019).
> > >
> > > [2] Yi, Kai, et al. "FedP3: Federated Personalized and Privacy-friendly Network Pruning under Model Heterogeneity." arXiv preprint arXiv:2404.09816 (2024).

---

> > > > ### Author Response · Authors · 2024-08-13
> > > >
> > > > We next discuss the baseline comparison. We first provide the comparison between FedP3 and our method by changing the global pruning rate from 0.5 and 0.9. We measure the uplink and downlink communication costs of each method. For FedP3, we used the OPU2 method with the ordered dropout pruning as our paper studies structured sparsity.
> > > >
> > > > | Algorithm    | Accuracy (CIFAR-100) | Uplink costs (Gbit) | Downlink costs (Gbit)| Total costs (Gbit) |
> > > > | -------- | ------- |------- |------- |------- |
> > > > | SpaFL  | $40.80\pm0.54$    | 2.3040 | 2.3040| 4.6080|
> > > > | FedP3 (0.5) | $37.73 \pm 0.42$     | 4.3853 | 2678.22| 2682.60 |
> > > > | FedP3 (0.7)    | $41.12 \pm 0.35$    |4.3853 | 3749.50| 3753.88 |
> > > > | FedP3 (0.9)    | $44.42 \pm  0.37$  | 4.3853 | 4824.73| 4825.17 |
> > > > |FedAvg|  $35.51\pm0.10$ |5361| 5361 |10712|
> > > >
> > > > Our method can achieve 35.38% of model density. We can see that our method outperforms FedP3 in terms of performance and communication costs at low model density (0.5). Although FedP3 performs better at high model density (0.7, 0.9), we note that our work focuses on optimizing sparse models with very low communication costs. We can see that the total communication costs of SpaFL is significantly less than FepdP3 (0.5). This means that SpaFL is more amenable to deployment on systems with limited resources. We also note that we only communicate trainable thresholds instead of doing full-layer aggregation of parameters.
> > > >
> > > > For the baselines, please note that Fjord (2021) and HeteroFL (2021) are still widely used for comparing the performance structured sparsity in FL (e.g., see [1] and [2]). While [2] studied personalized structured sparse models, [1] focused on global structured sparsity models. We will add [1] as a baseline in the revised version.
> > > >
> > > > [1] Dongping Liao, Xitong Gao, Yiren Zhao, and Cheng-Zhong Xu. Adaptive channel sparsity for
> > > > federated learning under system heterogeneity. In Proceedings of the IEEE/CVF Conference on
> > > > Computer Vision and Pattern Recognition, pages 20432–20441, 2023.
> > > >
> > > > [2] Chen, Daoyuan, et al. "Efficient personalized federated learning via sparse model-adaptation." International Conference on Machine Learning. PMLR, 2023.

---

> > > > > ### Comment · Reviewer_28Jj · 2024-08-13
> > > > >
> > > > > Thank you for your response and patience with my questions. I have two requests:
> > > > >
> > > > > 1. Could you please provide the explicit convergence rate and communication cost of your method, denoted in big-O notation?
> > > > > When compared with existing work, what insights can be drawn regarding the theory?
> > > > >
> > > > > 2. Can you elaborate on how the uplink, downlink, and total costs are computed? Why is the uplink always the same for FedP3 across different sparsity ratios?

---

> > > > > > ### Author Response · Authors · 2024-08-13
> > > > > >
> > > > > > Thank you for the constructive suggestions. We provide our response to each comment.
> > > > > >
> > > > > > >Q1: Could you please provide the explicit convergence rate and communication cost of your method, denoted in big-O notation? When compared with existing work, what insights can be drawn regarding the theory?
> > > > > >
> > > > > > To derive the explicit convergence rate in big-O notation, we follow the steps in [1] and [2]. We first assume $\eta = \frac{1}{\sqrt{T}}$ and $0 \leq \alpha < 1$. Then, we can bound $\frac{1}{\gamma(t)} \leq \frac{2}{\eta(t) (1 - \alpha)}$. By replacing $\eta(t)$ and $\gamma(t)$ with their assumed value and bound into the above convergence rate, we have the following bound.
> > > > > >
> > > > > > $\frac{1}{NT} \sum_{t=0}^{T-1} \mathbb{E} || \sum_{k=1}^N \nabla_{\tau} F_k (\tilde{w}_k(t), \tau(t)) ||^2 $
> > > > > > $\leq \frac{A}{ \sqrt{T} (1-\alpha)} + \frac{ B }{ T (1-\alpha)} + \frac{C}{\sqrt{T}} + \frac{D}{\sqrt{T}}$, where
> > > > > >
> > > > > >
> > > > > > $A = \sum_{t=0}^{T-1}  \sum_{k=1}^N \frac{\mathbb{E} ||  \nabla_{\tau} F_k (\tilde{w}\_k(t), \tau(t) ) - \nabla_{\tau_k} F_k ( \tilde{w}_k(t), \tau(t) ) ||^2}{MN}$,   $B =  \sum\_{t=0}^{T-1}  4\alpha || \exp(-\tau(t) ) ||^2$
> > > > > >
> > > > > > $ C =  M^2 n\_{in}^{max} G^2/2 $,  $ D = \sum_{t=0}^{T-1}  \sum_{k=1}^N \frac{\mathbb ||\tau(t) - \tau\_k(t)||^2 }{N}$.
> > > > > >
> > > > > > Therefore, the Big-O notation of the convergence rate becomes $\mathcal{O}(\frac{A}{ \sqrt{T} (1-\alpha)}) + \mathcal{O} ( \frac{ B }{ T (1-\alpha)}) + \mathcal{O}(\frac{C}{\sqrt{T}} ) + \mathcal{O} (\frac{D}{\sqrt{T}})$.
> > > > > >
> > > > > > To derive the communication cost/complexity [1], we need to bound the left-hand side in the form of $\frac{1}{NT} \sum_{t=0}^{T-1} \mathbb{E} || \sum_{k=1}^N \nabla_{\tau} F_k (\tilde{w}_k(t), \tau(t)) ||^2 \leq \epsilon$. To this end, we further bound the convergence rate using the dominant term $1/\sqrt{T}(1-\alpha)$ on the right-hand side as follows
> > > > > >
> > > > > > $\frac{1}{NT} \sum_{t=0}^{T-1} \mathbb{E} || \sum_{k=1}^N \nabla_{\tau} F_k (\tilde{w}_k(t), \tau(t)) ||^2 \leq \frac{D}{\sqrt{T}(1-\alpha)}$ , where $D = A +B +C +D$.  Then, to achieve $\epsilon$ stationary point, the communication complexity will be $\mathcal{O}(\frac{1}{\epsilon^2 ( 1-\alpha)^2})$. Since we only communicate thresholds at each round, the communication cost becomes $\mathcal{O} (\frac{ d\_{\tau} }{\epsilon^2 ( 1-\alpha)^2} )$, where $d\_{\tau}$ is the number of thresholds in a given model. We note that $d\_{\tau}$ is usually less than 1% of the number of parameters.
> > > > > >
> > > > > > From the above analysis, we can see that our convergence rate and communication complexity depends on the regularizer coefficient $\alpha$, which we used to enforce sparsity to models when updating thresholds. As $\alpha$ increases, it damages both the convergence rate and communication complexity. In the experiments, we used small values of $\alpha$ such as 0.001 or 0.00005 to balance the model sparsity and convergence rate (performance). Hence, our analysis shows how model sparsity can have impact on the convergence rate.
> > > > > >
> > > > > > >Q2: Can you elaborate on how the uplink, downlink, and total costs are computed?
> > > > > >
> > > > > > For the uplink, we measured the amount of parameters that clients sent to the server during the training. For the downlink, we measured the amount of parameters that the server sent to the clients during the training. The total cost is the sum of 'uplink cost' and 'downlink cost'. Then, we converted the number of communicated parameters to Gbit metric.
> > > > > >
> > > > > > >Q3: Why is the uplink always the same for FedP3 across different sparsity ratios?
> > > > > >
> > > > > > When calculating the uplink costs, we followed the implementation of FedP3. In their implementation, clients receive two sets of layers. The first set is a set of dense layers and another set is a set of pruned layers. According to their implementation, clients send the first set to the server, and this is why the uplink costs remain the same across different pruning ratios. In case, if we apply  pruning to the first set, then we have the following table,
> > > > > >
> > > > > > | Algorithm     | Uplink costs (Gbit) | Downlink costs (Gbit)| Total costs (Gbit) |
> > > > > > | -------- | ------- |------- |------- |
> > > > > > | SpaFL   | 2.3040 | 2.3040| 4.6080|
> > > > > > | FedP3 (0.5)    | 2.1923 | 2678.22| 2680.41 |
> > > > > > | FedP3 (0.7)    |3.0697 | 3749.50| 3752.57 |
> > > > > > | FedP3 (0.9)   | 3.9467 | 4824.73| 4828.67 |
> > > > > > |FedAvg|5361| 5361 |10712|
> > > > > >
> > > > > > We can still see that the our total costs are still smaller than that of FedP3.
> > > > > >
> > > > > > [1] Yi, Kai, et al. "FedP3: Federated Personalized and Privacy-friendly Network Pruning under Model Heterogeneity." arXiv preprint arXiv:2404.09816 (2024).
> > > > > >
> > > > > > [2] Yang, Haibo, et al. "Anarchic federated learning." International Conference on Machine Learning. PMLR, 2022.

---

> ### Comment · Reviewer_28Jj · 2024-08-14
>
> Thank you for your follow-up response. I appreciate the additional comparison with FedP3. I would suggest that the authors incorporate the points we've discussed into the paper, as this could significantly enhance both the clarity and contribution of the work. I also want to acknowledge the effort put into the rebuttal and the follow-up response.
>
> I would like to highlight a few important points:
>
> a) Assumption 3 regarding constant bounded gradient is quite strong and may not be feasible in the context of FL. It only seems reasonable when considering differential privacy to achieve privacy guarantees (perhaps future work on differential privacy could also help alleviate this.) This could be a potential limitation of your work, and it would be valuable to see a proper discussion of this in your future revisions. Additionally, directly assigning a complex term, such as A, B, etc., to the final convergence rate—especially when it depends on many variable factors, including the function itself—is not a reasonable way unless you have thoroughly examined its characteristics.
>
> b) The suggested theoretical comparison with existing methods is still missing. This will likely require substantial effort to carefully examine each variable in your results to ensure a fair comparison. Including this will greatly improve the soundness of the work.
>
> c) The calculation of uplink and downlink communication costs remains unclear. For example, you might follow Eqn (15) in [1] to provide an explicit computational analysis of the cost, which would be more convincing in demonstrating the potential benefits.
>
> d) While Fjord remains a popular method for comparison, I look forward to seeing comparisons with more recent approaches. I’m glad you included a comparison with FedP3 in the rebuttal, even though your focuses differ. If possible, I suggest incorporating additional comparisons with recent baselines, which would more convincingly demonstrate the merits of this work.
>
> Overall, while I believe there is substantial room for improvement in terms of clarity and completeness, I see this work as interesting and promising. I have decided to raise my score.
>
> [1] Malinovsky, Grigory, Kai Yi, and Peter Richtárik. "Variance reduced proxskip: Algorithm, theory and application to federated learning." Advances in Neural Information Processing Systems 35 (2022): 15176-15189.

---

> > ### Author Response · Authors · 2024-08-14
> >
> > We appreciate the reviewer for the constructive comments. We provide our response to each comment below.
> >
> > >Q1:  Assumption 3 regarding constant bounded gradient is quite strong and may not be feasible in the context of FL
> >
> > **A1:** In our Theorem 1, we derived the generalization guarantee through differential privacy. Specifically, we showed that the proposed algorithm satisfies differential privacy at each communication round by following [1]. Although our work does not focus on privacy, we will discuss about privacy guarantees in the revised version,
> >
> > >Q2: The suggested theoretical comparison with existing methods is still missing.
> >
> > **A2**: We compare our convergence analysis with [2], which provided the convergence rate of FL with arbitrary time-varying pruned models. Specifically, clients train pruned models, whose binary masks can change over time. and communicated pruned models with the server to generate a global model. The authors in [2] also made the same Assumption 1 and 3. Then, they further assumed that $||\theta_q - m_q \odot \theta_{q, n}||^2 \leq \delta^2 ||\theta_q||^2$, where $\theta_q$ is a global model at round $q$, $m_q$ is a binary mask, $0 \leq \delta < 1$ is pruning induced noise, and $\theta_{q, n}$ is the local model of client $n$. According to the Theorem in [2], they provide
> >
> > $\frac{1}{T} \sum_{t=1}^{T} \mathbb{E}  || \nabla F(\theta_q)||^2 \leq \frac{G_0}{\sqrt{T}} + \frac{V_0}{\sqrt{T}}  + \frac{H_0}{T} + \frac{I_0}{\Gamma\_{min}} \sum_{t=1}^{T} \mathbb{E} ||\theta_q||^2$, where $\Gamma\_{min}$  measures the minimum occurrence of the parameter in the local models in all rounds.
> >
> > Now we compare our convergence rate with the one from [2]. Although both algorithm converges to a stationary point with the rate of $1/\sqrt{T}$, we note that [2] has a non-vanishing term $\frac{\delta^2 I_0}{\Gamma\_{min}} \sum_{t=1}^{T} \mathbb{E} ||\theta_q||^2$ due to the noise from pruning. We also have similar terms such as  $\frac{\sum\_{t=0}^{T-1}  4\alpha || \exp(-\tau(t) ) ||^2}{T(1-\alpha)}$ due to the regularization. The main difference is that the convergence analysis of [2] cannot effectively capture the impact of sparse models due to uncontrollable and theoretical value $\delta$. Meanwhile, in our analysis, we can effectively show the impact of the regularization coefficient $\alpha$ on the convergence rate.
> >
> > >Q3: The calculation of uplink and downlink communication costs remains unclear.
> >
> > **A3:** We calculated the uplink and downlink communication costs from physically communicated amount of data between clients and the server during the fixed number of rounds. Specifically, each parameter can be represented with 32bit. For the uplink communication cost, we measured how many parameters are transmitted from clients to the server and then multiplied the number of transmitted parameters by 32 [bit] to derive the cost. We do a similar calculation for the downlink by measuring the number of parameters sent to clients. As the reviewer mentioned, we note that calculating the communication cost from the convergence rate can provide better understanding of the communication complexity as done in [3]. We will clarify how we calculated the up/down link communication costs and also provide the result by following [3] in the revised version.
> >
> > >Q4: While Fjord remains a popular method for comparison, I look forward to seeing comparisons with more recent approaches.
> >
> > **A4**: In the revised version, we will include the comparison with [4] and [5], which studied structured sparsity in FL.
> >
> > [1] Rong Dai, Li Shen, Fengxiang He, Xinmei Tian, and Dacheng Tao. Dispfl: Towards communication-efficient personalized federated learning via decentralized sparse training. In
> > International Conference on Machine Learning, pages 4587–4604. PMLR, 2022
> >
> > [2] Zhou, Hanhan, et al. "Every parameter matters: Ensuring the convergence of federated learning with dynamic heterogeneous models reduction." Advances in Neural Information Processing Systems 36 (2024).
> >
> > [3]  Malinovsky, Grigory, Kai Yi, and Peter Richtárik. "Variance reduced proxskip: Algorithm, theory and application to federated learning." Advances in Neural Information Processing Systems 35 (2022): 15176-15189.
> >
> > [4] Dongping Liao, Xitong Gao, Yiren Zhao, and Cheng-Zhong Xu. Adaptive channel sparsity for federated learning under system heterogeneity. In Proceedings of the IEEE/CVF Conference on Computer Vision and Pattern Recognition, pages 20432–20441, 2023.
> >
> > [5] Chen, Daoyuan, et al. "Efficient personalized federated learning via sparse model-adaptation." International Conference on Machine Learning. PMLR, 2023.

---

> > > ### Comment · Reviewer_28Jj · 2024-08-14
> > >
> > > Thanks for your response. I will keep my current score.

---

> ### Author Response · Authors · 2024-08-14
>
> We appreciate the reviewer for the constructive comments. We will update the convergence analysis, communication costs, and the comparisons with more recent baselines in the revision

---

### Comment · Area_Chair_Hn2U · 2024-08-14
**Thanks!**

Thanks to all reviewers for engaging with the authors!!

Reviewer 28Jj: I'd appreciate if you could read the latest response of the authors and reply to it.

Thanks!

AC

---

### Decision · Program_Chairs · 2024-09-25

**Decision:**

Accept (poster)

**Comment:**

This paper introduces SpaFL: a federated learning framework for enhancing communication efficiency and minimizing computational overhead by optimizing sparse model structures. This goal is achieved by defining a trainable threshold which leads to structured sparsity. Since the server and clients exchange thresholds only, SpaFL is able to reduce communication overhead. Furthermore, these trainable thresholds lead to lower computation overhead, too.

All reviewers see the paper favorably. In particular, all give acceptance-level scores, and list several compelling strengths, including:

- Communicating the threshold instead of the model weights is an interesting idea
- Preliminary empirical results on popular benchmarks seem to have shown that this idea indeed lowers communication overhead
- The paper is well-written and easy to follow

The several weaknesses that were pointed out by the reviewers were mostly addressed in the rebuttal and ensuing discussion. I fully expect these developments, clarifications and explanations to be included in the camera-ready version of the paper.